# Structure and mechanism of oxalate transporter OxlT in an oxalate-degrading bacterium in the gut microbiota

Titouan Jaunet-Lahary [1], Tatsuro Shimamura [2] ✉, Masahiro Hayashi[3], Norimichi Nomura [2], Kouta Hirasawa[2], Tetsuya Shimizu[4], Masao Yamashita[4], Naotaka Tsutsumi [3,5], Yuta Suehiro [5], Keiichi Kojima [3], Yuki Sudo [3], Takashi Tamura [6], Hiroko Iwanari[7], Takao Hamakubo [7], So Iwata[2], Kei-ichi Okazaki [1] ✉, Teruhisa Hirai [4] ✉ & Atsuko Yamashita [3,4,5] ✉

An oxalate-degrading bacterium in the gut microbiota absorbs food-derived oxalate to use this as a carbon and energy source, thereby reducing the risk of kidney stone formation in host animals. The bacterial oxalate transporter OxlT selectively uptakes oxalate from the gut to bacterial cells with a strict discrimination from other nutrient carboxylates. Here, we present crystal structures of oxalate-bound and ligand-free OxlT in two distinct conformations, occluded and outward-facing states. The ligand-binding pocket contains basic residues that form salt bridges with oxalate while preventing the conformational switch to the occluded state without an acidic substrate. The occluded pocket can accommodate oxalate but not larger dicarboxylates, such as metabolic intermediates. The permeation pathways from the pocket are completely blocked by extensive interdomain interactions, which can be opened solely by a flip of a single side chain neighbouring the substrate. This study shows the structural basis underlying metabolic interactions enabling favourable symbiosis.

Oxalate is the smallest dicarboxylate ($C_2O_4{}^{2-}$) ingested through our daily diet from oxalate-containing foods[1], such as vegetables, beans and nuts[2]. Oxalate is also a final metabolic product in our body and is partly secreted to the intestine via the systemic circulation[1]. Then it is absorbed from the intestinal tract and excreted through the kidney[3]. However, excess oxalate forms an insoluble salt with blood calcium and causes kidney stone disease (Fig. 1a). *Oxalobacter formigenes* is an oxalate-degrading bacteria in the gut[4] that can metabolically decomposes intestinal oxalate and thus contributes significantly to oxalate homeostasis in the host animals including humans[3,5,6]. Indeed, patients

with cystic fibrosis[7] or inflammatory bowel disease[8] or those who have undergone jejunoileal bypass surgery[9] are known to have low rates of colonisation of *O. formigenes* and an increased risk of hyperoxaluria and kidney stone formation.

Oxalate transporter (OxlT), an oxalate:formate antiporter (OFA)[10] in *O. formigenes*, is a key molecule for oxalate metabolism in this bacterium. OxlT catalyses antiport of carboxylates across the cell membrane according to their electrochemical gradients with a substrate specificity optimised to the C2 dicarboxylate, oxalate. Indeed, the transporter shows a high turnover rate (>1000/s) for oxalate self-

[1]Research Center for Computational Science, Institute for Molecular Science, National Institutes of Natural Sciences, Okazaki 444-8585, Japan. [2]Graduate School of Medicine, Kyoto University, Kyoto 606-8501, Japan. [3]Graduate School of Medicine, Dentistry and Pharmaceutical Sciences, Okayama University, Okayama 700-8530, Japan. [4]RIKEN SPring-8 Center, Sayo 679-5148, Japan. [5]School of Pharmaceutical Sciences, Okayama University, Okayama 700-8530, Japan. [6]Graduate School of Environmental and Life Sciences, Okayama University, Okayama 700-8530, Japan. [7]Research Center for Advanced Science and Technology, The University of Tokyo, Tokyo 153-8904, Japan. ✉e-mail: t.shimamura@mfour.med.kyoto-u.ac.jp; keokazaki@ims.ac.jp; teruhisahirai@gmail.com; a_yama@okayama-u.ac.jp

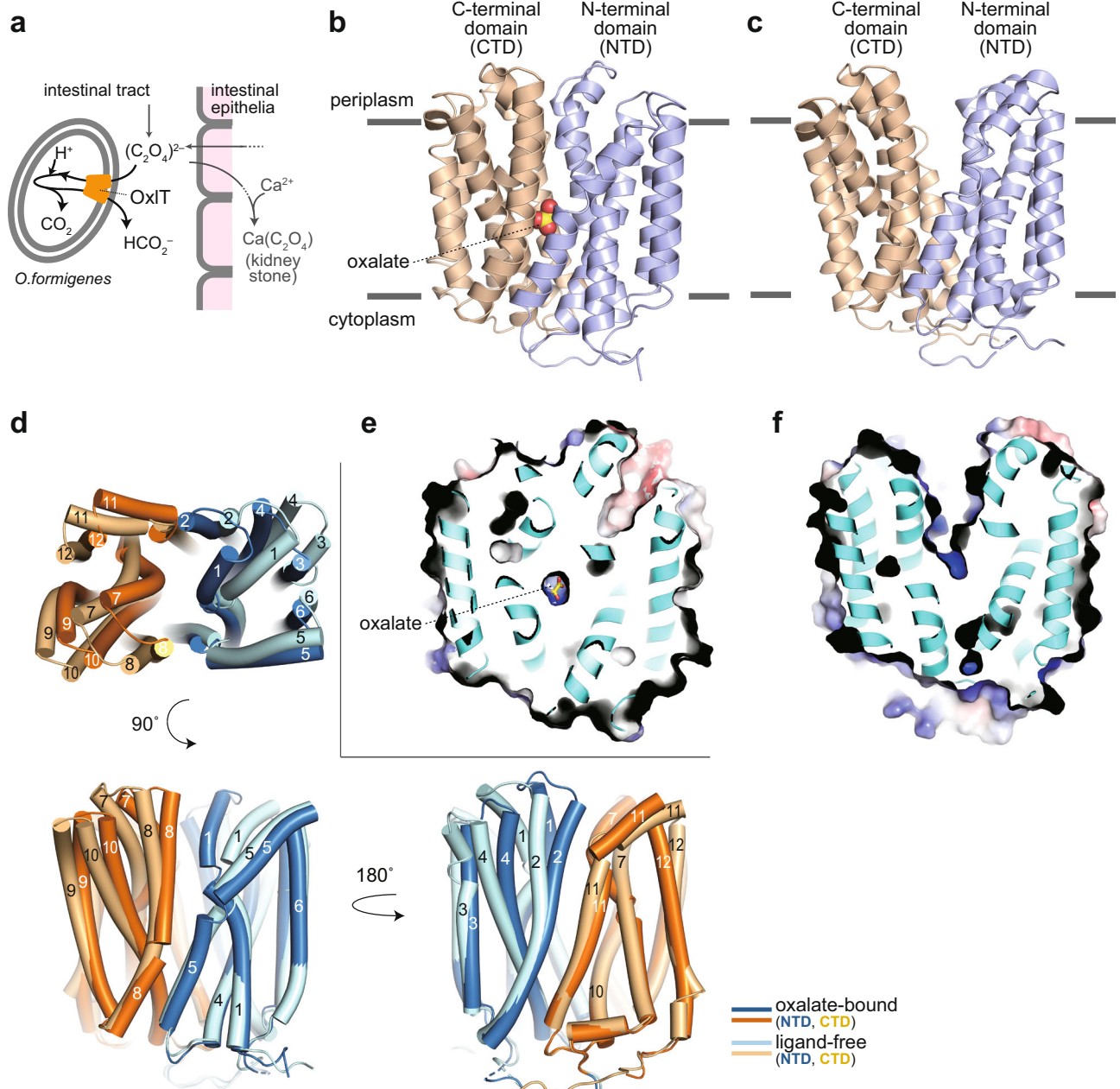

**Fig. 1 | Structure of OxlT. a** Schematic drawing of OxlT function in the oxalate-degrading bacterium, *O. formigenes*, in the gut. **b, c** Crystal structures of the oxalate-bound (PDB ID 8HPK; **b**) and ligand-free (PDB ID 8HPJ; **c**) OxlT. **d** Superposition of oxalate-bound and ligand-free OxlT. A view from the periplasm (top) and two views in the transmembrane plane (bottom) are shown. **e, f** Surface electrostatic potential map of oxalate-bound (**e**) and ligand-free (**f**) OxlT. Electrostatic potentials at ±5 kTe⁻¹ were mapped on the surfaces.

exchange[11,12]. Under physiological conditions in the oxalate autotroph *O. formigenes*, the carboxylate-exchange function of OxlT enables uptake of oxalate from the host intestine as a sole carbon source for the bacterium and a release of formate ($HCO_2^-$), the final degradation product of oxalate that is toxic if accumulated in the bacterial cell[11–13] (Fig. 1a). OxlT catalytic turnover of the oxalate:formate exchange is accompanied by the metabolic degradation of oxalate to formate via a decarboxylase that consumes a proton in the cytosol, consequently producing a proton electrochemical gradient across the bacterial cell membrane[11]. Therefore, OxlT serves as a 'virtual proton pump' that creates a proton motive force for bacterial ATP synthesis[11]. Thus, the functional characteristics of OxlT as an antiporter between oxalate and formate, rather than a uniporter of each chemical, is essential to couple carbon metabolism and energy formation. Notably, OxlT does

not accept oxaloacetate ($C_4H_2O_5^{2-}$) or succinate ($C_4H_4O_4^{2-}$), which are Krebs cycle dicarboxylate intermediates, as substrates[13]. These dicarboxylates with four carbon atoms (C4 dicarboxylates) are important metabolic intermediates at the bacterial cytosolic side while they are also absorbed as energy sources and biosynthetic precursors through an intestinal transporter at the host lumen side[14]. Therefore, the ability of OxlT to discriminate between C2 and C4 dicarboxylates is critical for the favourable symbiosis between host animals and the gut bacterium.

OxlT belongs to the major facilitator superfamily (MFS), the large transporter family whose members transport a wide array of chemicals[10]. MFS proteins share a common architecture of twelve transmembrane (TM) helices that contain symmetrical N- and C-terminal halves of six gene-duplicated TM units, with a substrate-binding site in the centre of the molecule[15,16]. The substrate transport

mechanism of the MFS family as well as other transporter families, is explained by the 'alternating access model'[17], whereby transporter molecules open a cavity from the binding site to either side of the membrane alternately, and take outward-facing, occluded and inward-facing conformations via a 'rocker switch' motion of the N- and C-terminal domains, thereby allowing substrate transfer across the membrane[18–20]. Although a wealth of structural information of each MFS member has been accumulated, current knowledge about the OFA family remains limited to an OxlT structure initially solved by electron crystallography at 6.5 Å[21,22]. Therefore, the specific oxalate recognition and antiport mechanism of OxlT is yet to be elucidated in a higher resolution structure.

In this study, we report the X-ray crystallographic structures of OxlT in oxalate-bound and ligand-free forms solved at 3.0–3.3 Å to understand the structural basis of these key transporter functions that underly the symbiosis of this oxalate-degrading bacterium in the gut.

## Results

### OxlT structures in two different conformations

The wild-type OxlT is unstable under various conditions, such as in the presence of chloride ion[12,23], which significantly narrows the available chemical space for crystallisation screening. For the structural study of OxlT, an antibody-assisted crystallization strategy was adopted. In general, Fab or Fv antibody fragments that bind specifically to conformational epitopes in membrane proteins can increase the hydrophilic surface area available for the formation of rigid crystal lattices. Additionally, the bound antibody fragments can reduce the inherent protein flexibility and conformational heterogeneity, increasing the likelihood of successful crystallization of membrane proteins[24,25]. OxlT was stabilised by binding two different antibody fragments, resulting in crystallisation under two different conditions. We confirmed that the antibody fragments used for crystallisation bind to OxlT both in the presence and absence of oxalate (Supplementary Fig. 1). Therefore, it is unlikely that the antibody fragments trap OxlT in a particular conformation artificially. The crystal structure of oxalate-bound OxlT in complex with the Fab fragment was solved at 3.0 Å (PDB ID 8HPK) while that of ligand-free OxlT in complex with an Fv fragment was solved at 3.3 Å (PDB ID 8HPJ) (Supplementary Fig. 2a, b and Supplementary Tables 1 and 2).

The overall structure of OxlT consists of 12 TM helices (Fig. 1b, c), as observed in the previous EM structure[21] and later confirmed as a typical MFS architecture[15,16]. In the oxalate-bound state, the OxlT protein adopts an occluded conformation with an oxalate molecule binding at the centre of the structure (Fig. 1b, e). In contrast, the ligand-free OxlT takes a substantially different conformation from the oxalate-bound form (Fig. 1c, d, f). The OxlT protein displays a large V-shaped cavity between the N- (TM1–6) and the C-terminal (TM7–12) domains, which is connected from the central oxalate binding site to the periplasm, a clear signature of an outward-facing conformation.

In a comparison of the occluded and outward-facing structures, the Cα root-mean-square-deviation (RMSD) for all residues was 2.6–2.7 Å (Fig. 1d). Even the sole N- or C-terminal domains of the two showed considerable structural differences (Cα RMSD of -1.5–1.6 Å). Therefore, the structural change between the occluded and outward-facing states with a 'rocker switch' motion is not achieved by the tilt of the rigid structural units but is concomitant with their bending. Indeed, conspicuous bends at the periplasmic portion are observed on TM1, 2, 4, 7, 8, and 11 in the outward-facing structure, with tilting of the other surrounding TM helices (Fig. 1d). In contrast, there was no notable differences in the cytoplasmic portion between the two conformations. In other MFS proteins, such as GLUT5 and NarK, bending at the glycine residues in the TM helices has been observed between the different conformational states[26,27]. OxlT has 52 glycine residues, which is approximately one eighth (12.4%) of the amino acid content (Fig. 1d, Supplementary Figs. 2c, d and 3). Notably, this glycine

frequency is higher than that in other MFS proteins, such as LacY (8.6%), GLUT5 (7.6%) or NarK (10.4%), and in TM helices in other membrane proteins (-8.7%)[28]. Therefore, the accumulation of bending of the TM helices at the glycine residues appears more prominent in achieving the conformational switch between the states in OxlT. Glycine residues were also found at the interface between the N- and C-terminal domains as in TM5 and TM8 or TM2 and TM11 (Supplementary Fig. 2c, d) and achieved tight helical packing as previously reported[28,29]. The high glycine occurrence observed in OxlT may be required to occlude the oxalate, which is small for a transported substrate, in the centre of the molecule. Conversely, the glycine-rich architecture is likely responsible for the instability of OxlT in detergent micelles, which hinders crystallization in the absence of antibody fragments and functional assays, as described below. The high glycine ratio was also observed in the other OFA proteins (10.2 ± 1.1% with 15 strictly conserved positions; observed in 11 family members shown in Supplementary Fig. 3) and may be a family trait.

### Oxalate-bound occluded structure

In the crystal structure (PDB ID 8HPK), the oxalate molecule binding to OxlT is refined as a twisted configuration (Fig. 2a and Supplementary Fig. 4a). The bond between the two carboxyl groups in an oxalate dianion is known to be a single and unconjugated, allowing a free rotation of the carboxyl groups about the C-C bond[30]. Since the resolution of the oxalate-bound OxlT structure is insufficient to precisely determine the dihedral angle of oxalate, we performed quantum mechanics (QM) and quantum mechanics/molecular mechanics (QM/MM) calculations of the oxalate binding in the occluded OxlT structure to examine the energetically minimised conformation. The resulting O-C-C-O dihedral angles in the oxalate were within 50–68˚ (Supplementary Fig. 4b, c and Supplementary Table 3). These values are close to those observed in the original crystal structure (60.1˚), verifying that the oxalate is not planar but twisted in the crystal structure.

At the binding site in OxlT, oxalate binds to the transporter with one carboxyl group forming a bidentate salt bridge with Arg272 in TM8 while the other forms an ionic interaction with Lys355 in TM11 (Fig. 2a). In addition to the salt-bridging with the oxalate, the ε-amino group in Lys355 forms an interdomain hydrogen bond network with the carboxamide groups in Gln34 (TM1) and Gln63 (TM2) in the N-terminal domain. Similarly, the guanidino group in Arg272 forms an interdomain hydrogen bond with the main chain carbonyl group in Ala147 (TM5) and further interacts with the side-chain carboxamide and main-chain carbonyl group of Asn268 upstream of TM8. The region around Arg272 is the bending point in TM8 due to the sequence of N[268]GGCR[272]P, and therefore the hydrogen bonds between Arg272 and Asn268 likely maintain the conformation and orientation of TM8 in the oxalate-bound structure. These inter- and intra-domain hydrogen bonding networks involving Arg272 and Lys355 likely play pivotal roles in organising the structure of the binding pocket and stabilising the occluded conformation, despite the location of these two basic residues within the C-terminal domain. These two basic residues are conserved within the OFA family (Supplementary Fig. 3), are essential for oxalate transport, and even R272K or K355R mutations reduce transport activity[31–33]. These results confirm observations that not only the charges but also the chemical structures of the side-chains of the two residues are important for the structural organisation of the binding site.

In addition to the two basic residues, numerous aromatic residues are found to contribute to oxalate binding. The hydroxyl groups of Tyr35 and Tyr124 form hydrogen bonds with either of the carboxyl groups in oxalate (Fig. 2a). Furthermore, the aromatic side chain groups in Tyr150, Trp324, Tyr328 and Trp352 form face-to-face or edge-to-face π-π interactions with the carboxyl groups in oxalate, indicating the significance of the π-electron systems in oxalate for molecular recognition by OxlT. These aromatic residues distributed in

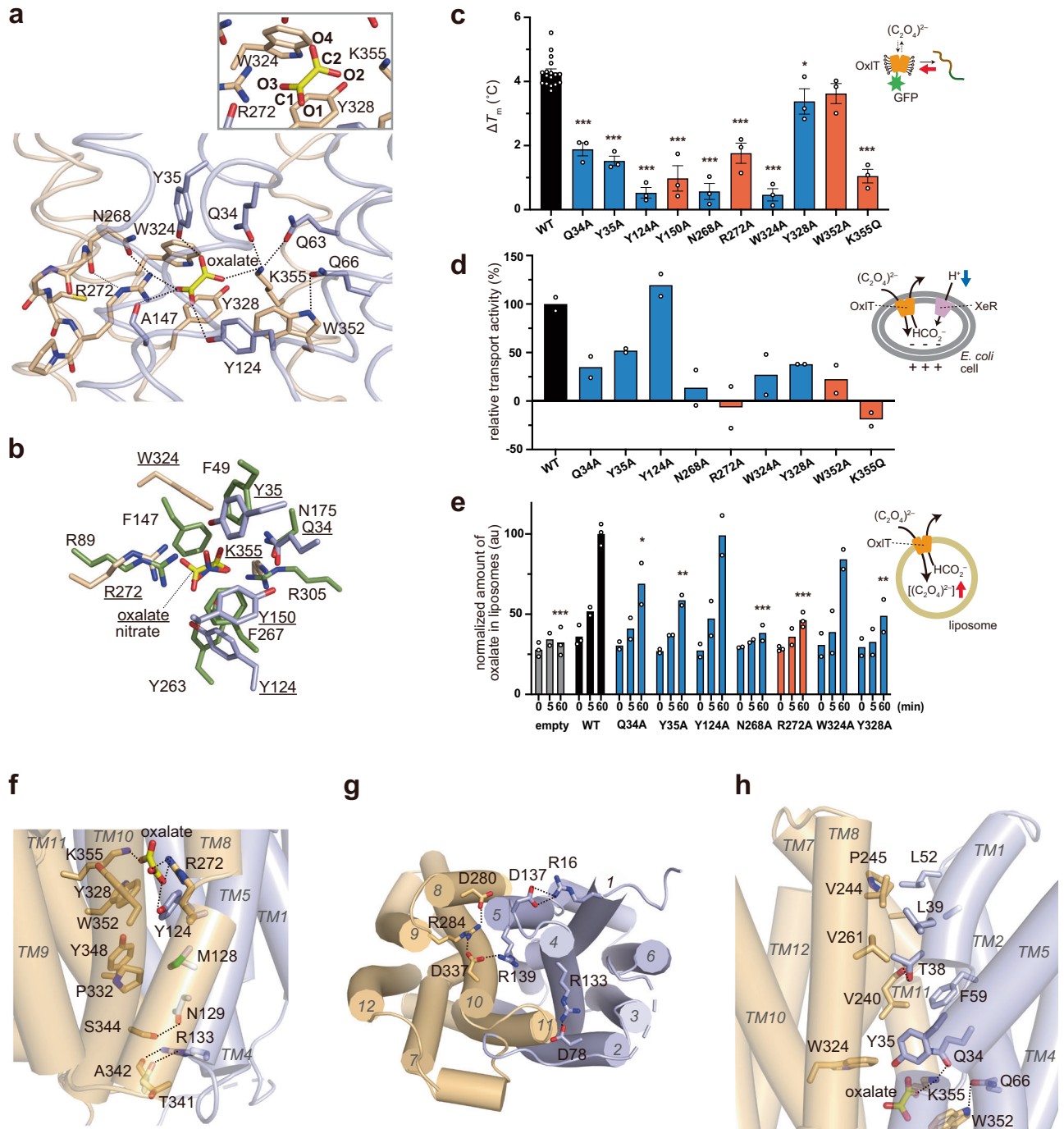

**Fig. 2 | Oxalate-bound occluded OxlT structure. a** Close-up of the binding site in oxalate-bound OxlT (PDB ID: 8HPK). Dashed lines indicate potential hydrogen bonds and salt bridges. Also shown a magnified view of the bound oxalate molecule with atom labels. **b** Superposition of the substrate-binding site structures of OxlT (with the underlined labels) and NarK (green, with normal labels; PDB ID: 4U4W) based on the topological similarity of the amino acid residues interacting with the substrates. **c** Oxalate binding assay by GFP-TS. Data represent means ± SEM of the increases in melting temperatures caused by the addition of 3 mM potassium oxalate in three independent experiments. WT: wild-type. **d** Oxalate uptake assay using recombinant *E. coli* cells. The relative transport activities of the mutant OxlT to that of the WT OxlT measured on the same day set as 100%, while that of the cells without expressing OxlT[36] was set as 0%, are displayed. The bars represent the means of the technically duplicated measurements from a single experiment for

each mutant. The results of the R272A, and K355Q mutants are reposted from the previous study[36] for comparison. **e** Proteoliposome oxalate uptake assay. The resultant oxalate concentrations in liposome lysates were measured and normalised to those of the WT OxlT at 60 min. The results of the liposomes without OxlT are shown as "empty." The bars represent the means of the results in three (data at 0 min and 60 min for empty, WT, and R272A) or two (others) independent experiments. au: arbitrary unit. In panels **c** and **e**, the data were analysed by a two-sided one-way analysis of variance with Dunnett's test with the WT OxlT as a control, and the exact *P* values are provided in Supplementary Table 6. *$P < 0.05$, **$P < 0.01$, ***$P < 0.001$. The red bars denote the mutants (or mutations at the same residue) exhibiting a loss of activity in previous studies[31,32,34]. **f** Interdomain interactions closing the cavity to cytoplasm. **g** Ionic interaction network at the cytoplasmic side of OxlT. **h** Interdomain interactions closing the cavity to periplasm.

both the N- and C-terminal halves, and thus their interactions with oxalate also stabilise the closure of the interdomain cavities to achieve the occluded conformation. In addition, Trp352 (TM11) forms an interdomain hydrogen bond with Gln66 (TM2). These aromatic residues are conserved among proteins belonging to the OFA family (Supplementary Fig. 3). Indeed, mutations at Tyr150 or Trp352 have been linked to the loss of transport functions in previous studies[31,34]. Notably, a similar combination of ionic and π-π interactions was observed with the recognition of nitrate, which also has a π-electron system, by NarK in the nitrate/nitrite porter (NNP) family[26,29], although the NNP family is distant from the OFA family[10] and the positions of the involving residues do not correspond to each other (Fig. 2b).

The significance of the above-mentioned or their neighbouring residues for oxalate binding and transport was investigated. Since the purified OxlT is unstable, particularly in the absence of substrate, we utilised a variety of functional assays: (1) the GFP thermal shift (GFP-TS) assay[35] to assess the ability of oxalate binding, utilising the crude, detergent-solubilized OxlT-GFP fusion protein containing lipids from the host *E. coli* cells; (2) an in cellulo transport assay using *E. coli* cells recombinantly expressing OxlT[36]; and (3) an in vitro transport assay using proteoliposomes reconstituted with the purified OxlT. In the GFP-TS assay, wild-type OxlT exhibited thermal stabilisation upon oxalate binding. In contrast, the extent of oxalate-dependent thermal stabilisation was reduced in the mutant OxlT proteins Q34A, Y35A, Y124A, Y150A, N268A, R272A, W324A and K355Q (Fig. 2c). Since the thermostabilities of the mutant OxlT proteins were not significantly different from that of the wild-type OxlT in the absence of oxalate (Supplementary Fig. 5), these results suggested that the mutation at the residues resulted in reduced binding affinity of oxalate and/or reduced stability of the bound structure. In the in cellulo transport assay, the extent of oxalate–formate exchange by OxlT in *E. coli* cells, which is negatively electrogenic, was assessed by coupled light-driven inward proton transfer by a co-expressed xenorhodopsin and the resultant pH increase of the external solution[36] (Fig. 2d). In addition to non-functional mutants R272A and K355Q[31,32], which had been verified their loss of activity in our previous study[36], OxlT mutations with Q34A, Y35A, N268A, W324A, Y328A and W352A also reduced activity (Fig. 2d). Finally, the in vitro transport assay using the purified OxlT reconstituted into proteoliposomes was performed for R272A and the mutants which have not been tested by this method in the previous studies (Fig. 2e and Supplementary Fig. 6). A similar trend was observed between the proteoliposome and in cellulo assays, despite some small deviations due to the differences in the two systems. Intriguingly, while most mutants demonstrated a decrease in both binding and transport activities, Y124A and W324A mutations had no significant effects on transport activities, at least in either of transport assays, despite the reduced oxalate binding capacity. These observations may be explained by the fact that the lowering the affinity can result in acceleration of the net uptake due to rate-affinity trade-off and/or reduced substrate efflux[37]. In contrast, Y328A and W352A showed similar oxalate binding but reduced oxalate uptake activities compared to the wild-type OxlT, indicating the importance of these residues on the catalytic turnover. Therefore, the results from the functional studies suggested that the residues in the vicinity of bound oxalate in the OxlT structure play significant roles in oxalate binding and/or transport.

The interactions between the substrate and the residues in the occluded OxlT crystal structure are optimised to the C2 dicarboxylate oxalate. Since the oxalate molecule tightly fits to the binding pocket, replacing oxalate with a larger dicarboxylate, such as Krebs cycle intermediates, causes steric clashes with the residues in OxlT, likely destabilising the occluded conformation (Supplementary Fig. 7a). A flexible docking study resulted in a position that could accommodate a C3 dicarboxylate, malonate, in the binding site of the occluded OxlT, although this had fewer interactions compared with the case for

oxalate, due to the rearrangement of amino acid residues in the binding pocket (Supplementary Fig. 7b). This is consistent with the decreased affinity and transport activity for malonate with a $K_d$ value of 1.2 mM compared to 0.02 mM for oxalate[13]. Even with flexible docking, no pose for the binding of C4 dicarboxylates to the occluded OxlT was observed, consistent with a previous report indicating that these molecules do not compete significantly with oxalate for uptake by OxlT[13]. The GFP-TS assay confirmed the highest specificity of oxalate to OxlT; the thermal stabilisation of OxlT was observed prominently in the presence of oxalate (C2) but not in the presence of malonate (C3) or succinate (C4) (Supplementary Fig. 7c).

From the oxalate binding site to the cytoplasm or periplasm, extensive intramolecular interactions were observed between TM helices in the N- and the C-terminal domains, such as TM2 and TM11, TM5 and TM8, the periplasmic halves of TM1 and TM7, and the cytoplasmic halves of TM4 and TM10 (Fig. 2f–h). These interactions stabilise the closure of the interdomain cavities in the occluded structure.

In the cytoplasmic side below the oxalate binding site, hydrophobic interactions involving Met128 (TM4), Pro332 (TM10) and Tyr348 (TM11) were observed, followed by polar interactions between Asn129 (TM4) and Ser344 (TM11), Arg133 (TM4) and the main chain carbonyl groups of Thr341 and Ala342 (TM11) (Fig. 2f). These interactions are further supported with by charge-dipole interactions at the cytoplasmic end, formed between Asp78 (TM2) or Asp280 (TM8) and the N-terminal ends of TM11 or TM5, respectively (Fig. 2g). The two aspartate residues are located in the 'A-like' motifs in the TM2-3 ('G74YFVD78KFGP82R83IP' sequence, A^{L2-3}) or TM8-9 (G276FVSD280KI-GR284YK, sequence, A^{L8-9}) regions (Supplementary Fig. 3). Motif A is one of the commonly conserved motifs in MFS proteins, and the D(+5) is known to participate in an interdomain charge-helix dipole interactions[38]. Notably, these aspartate residues further compose extensive ionic interaction networks in the cytoplasmic side (Fig. 2g). Specifically, Asp78 and Arg133 in TM4, and the downstream residue Asp137 and Arg16 in TM1, form salt bridges. Further downstream, Arg139 at the N-terminal end of TM5 forms a charge relay network with Asp337, Arg284 and Asp280.

In the periplasmic side above the oxalate binding site, a hydrogen bond between Thr38 (side chain) in TM1 and Val240 (backbone) in TM7 (2.72 Å) closes the pore tunnel in the occluded conformation (Fig. 2h). Above the hydrogen bond, Leu39 (TM1), Leu52 (TM2), Val244 and Pro245 (TM7) and Val261 (TM8) form hydrophobic interactions.

### Ligand-free outward-facing structure
In contrast to the occluded substrate-binding site in the oxalate-bound OxlT, a large cavity from the binding site to the periplasmic space is open in the ligand-free OxlT (PDB ID 8HPJ; Fig. 1f). At the empty binding site, the Lys355 side chain flips out from Arg272 due to charge repulsion and shifts the positions from those found in the oxalate-bound form (Fig. 3a). In the ligand-free form, most of the interdomain hydrogen bonds observed in the oxalate-bound state are retained. However, the one between Lys355 in the C-terminal domain and Gln34 in the N-terminal domain is likely disrupted in the ligand-free state, judging by the distance between the side chains (>4 Å). Positional shifts of the surrounding aromatic residues, such as Tyr35, Tyr150, Trp324 and Tyr328, were also observed (Fig. 3b). These changes at the substrate-binding site due to the absence of oxalate likely underlie the structural rearrangement of the overall architecture and result in the conformational change between the occluded and outward-facing state. Notably, the cavity opening to the periplasm displayed an extensive positively charged surface (Figs. 1f and 3c). This basic property is mainly derived from Arg272 and Lys355 in the binding site. In addition, the side-chain amino groups in Lys45 and Arg248 and the amide groups in Gln34, Asn42, Gln56, Gln264, Asn265 and Asn268, that line this cavity, are now exposed to the solvent. These groups and the positive dipole moments of the bent helices of TM1, TM5 and TM11 also

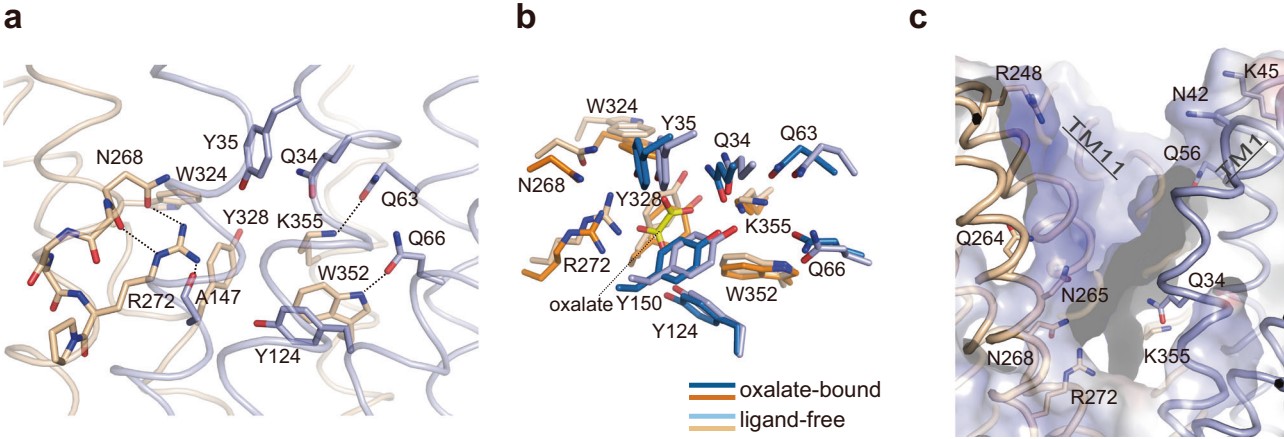

**Fig. 3 | Ligand-free outward-facing OxlT structure. a** Close up of the binding site in ligand-free OxlT (PDB ID: 8HPJ) viewed from the same orientation in Fig. 2a. The domain colour coding is as in Fig. 1c. The dashed lines indicate potential hydrogen bonds. **b** Superposition of the substrate-binding site structures of OxlT in oxalate-bound and ligand-free forms. **c** Close up of the cavity open to the periplasm. Models of polar residues exposed to the cavity and the surface coloured with the electrostatic potential map at ±5 kTe⁻¹ are also shown. In panels **a**–**c**, the molecule defined as chain A is shown as a representative.

contribute to the basic property of the entire cavity (Fig. 3c). The charge repulsion caused by Arg272 and Lys355 at the empty ligand-binding site as well as the extensive basic surface of the cavity likely prevents closure of the pocket to the occluded form in the absence of oxalate, thus stabilising an open state. The stability of an open state conformation in the absence of a substrate, which prevents transition to the occluded state, underlies the OxlT function as an antiporter, in which the conformational switch in the absence of a substrate during the catalytic process is disallowed[22,38]. A similar situation was observed on a nitrate/nitrite antiporter NarK[29], where the positively charged surface of the open cavity stabilised the inward-facing conformation[26].

On the other hand, the cytoplasmic part of the ligand-free OxlT structure shows no significant changes from that of the oxalate-bound structure (Fig. 1d).

## Substrate-binding and conformational dynamics of OxlT

To address the structural dynamics of OxlT enabling the conformational switch necessary for the transport cycle, we performed molecular dynamics (MD) simulations[39] based on the oxalate-bound occluded and the ligand-free outward-facing OxlT crystal structures.

We first simulated oxalate binding to the ligand-free outward-facing conformation (PDB ID: 8HPJ) by positioning the oxalate outside of the protein (Fig. 4a–c and Supplementary Fig. 8). At Gln34, Tyr35, Arg272, Tyr328 and Lys355, oxalate ion binding to the binding site of OxlT was observed (Fig. 4b). The distance between the geometric centres of the oxalate ion and the binding site residues, with a cutoff distance of 5 Å, determined the binding (Fig. 4c). The interaction of oxalate ion with Lys355 resolved its charge repulsion with Arg272 in the ligand-free form and restored the side chain's configuration from the flipped state. The rapid binding of the negatively charged oxalate ion is facilitated by an extensive positively charged surface (Figs. 1f and 3c). The stability of the bound conformation depended on the protonation state of Lys355 (see Methods for p$K_a$ calculation), which can be influenced by the luminal pH in the gut, which varies from 5 to 8 by region[40], and the hydrophobic environment in the binding site[41]. For the protonated Lys355, a single binding event was observed, and the bound oxalate ion mostly remained in the binding site for the remainder of the simulation (grey line in the top panel of Fig. 4c). In contrast, for the neutral Lys355, multiple binding and unbinding events of different oxalate ions were observed (coloured lines in the bottom panel of Fig. 4c). This results in a higher occupancy rate of 98.6% for the protonated Lys355 than that of 77.0% for the neutral Lys355, which is calculated by the above definition of the bound state using the oxalate

and binding site distance. Thus, a lower oxalate dissociation constant is predicted for the protonated Lys355. During the 1.7 μs simulations, the outward-facing conformation of OxlT was stable, as shown in the plot of the RMSD of the backbone atoms from the outward-facing crystal structure (Fig. 4a). The results suggest that the binding of the oxalate observed in the simulations is an early-stage binding mode that should be followed by the conformational rearrangement and desolvation of the binding site and the transition to the occluded conformation to adapt the fully bound conformation.

We next addressed the conformational dynamics of the occluded conformation (PDB ID 8HPK) in the oxalate-bound state. During the 1 μs simulation, two out of three independent trajectories remained in the occluded state (Fig. 4d). In the occluded conformation, most water molecules were blocked at certain positions in the periplasmic and cytoplasmic sides of the transporter during the simulations, although some entered OxlT (Fig. 4e and Supplementary Movie 1). A water density analysis pinpointed structural layers blocking entry of water into the oxalate binding site during the simulation (Supplementary Fig. 9). One of these is a hydrophobic layer consisting of Thr38 and Leu39 in TM1, Val244 in TM7 and Val261 in TM8 at the periplasmic side (lower left panel of Fig. 4e). This layer, combined with the hydrogen bond between Thr38 and Val240 in TM7 (shown by a broken line in lower left panel of Fig. 4e), also blocked the exit of ligand to the extracellular side and thus served as the periplasmic gate. The other layer consists of Met128 in TM4, Pro332 in TM10 and Tyr348 in TM11 at the cytoplasmic side (lower right panel of Fig. 4e). These periplasmic and cytoplasmic hydrophobic gates, together with the TM1–TM7 hydrogen bond, have similarity with the previously reported NarK transporter[42], based on residues located at similar positions to those in OxlT in the aligned structure (Supplementary Fig. 10). This result suggests that the hydrophobic gates[39] are a conserved mechanism among the two transporters.

In contrast, in one trajectory from the occluded conformation, an opening of the periplasmic gate was observed (blue line in Fig. 4d). In the transition, the flip of Gln34 side chain from the binding site occurred first (Fig. 4f, Supplementary Fig. 11a and Supplementary Movie 2). The Gln34 flip resulted in a disruption of the hydrogen bond with Lys355, as observed in the outward-facing crystal structure (Fig. 3a). Furthermore, since Gln34 is located one-turn upstream of Thr38 in TM1, the flip also caused a constant disruption of the hydrogen bond between Thr38 and Val240, which was bonded on and off by thermal fluctuation even before the flip (shown by a broken line in Fig. 4f and Supplementary Fig. 11b). After ~280 ns following the

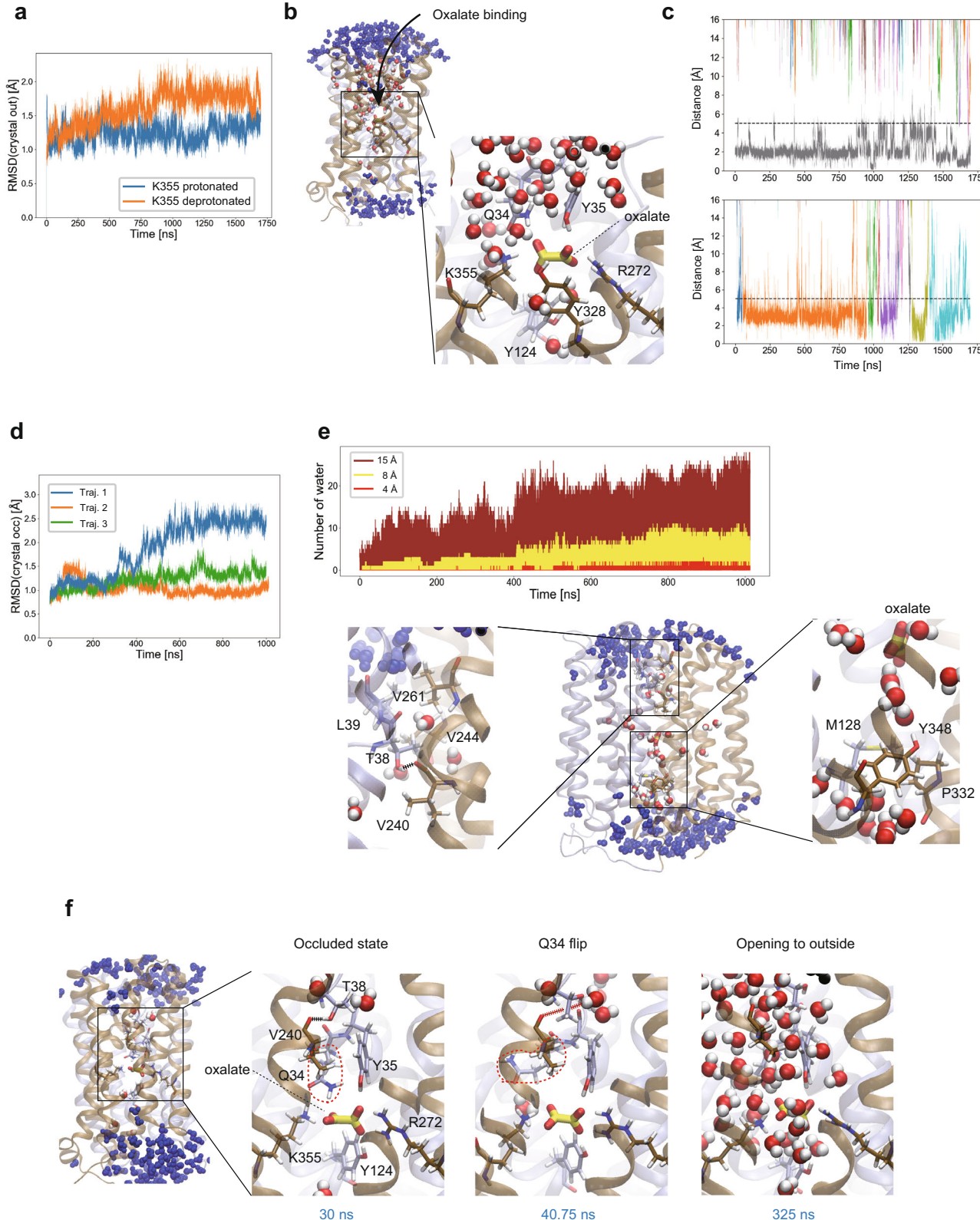

Gln34 flip, OxlT started opening to the outside and many water molecules entered the transporter (Fig. 4f and Supplementary Movie 2). We note that the Gln34 flip is a transient conformation and that the side chain returned to the original position after reaching the outward-facing state in the last part of the simulation, consistent with the observation on the outward-facing crystal structure (Supplementary Fig. 11a, c). Notably, the Gln34 flip was also observed in a trajectory

starting from the occluded conformation with formate modelled in the binding site (Supplementary Fig. 12). In this trajectory, the hydrogen bond between Thr38 and Val240 was again completely broken after the Gln34 flip, followed by a transition from the occluded to the outward-facing conformation, in accordance with the physiological reaction of the formate release to periplasm in *O. formigenes*. In contrast, the Gln34 flip was not observed in any of the other trajectories

**Fig. 4 | Substrate binding and conformational dynamics of OxlT. a–c** MD simulations started from the ligand-free outward-facing OxlT crystal structure (PDB ID 8HPJ). **a** The RMSDs from the initial outward-facing crystal structure are shown for two trajectories with different protonation states of Lys355 in different colours. **b** Snapshot of the bound oxalate to OxlT with protonated Lys355. In the zoom-out snapshot, water molecules within 15 Å distance from the oxalate ion are shown in the CPK colour, while those between 15 and 25 Å distance are shown in blue. In the close-up snapshot, water molecules within 15 Å distance from the oxalate ion are shown. **c** Distances of the oxalate ions from the binding site in a single trajectory, either with protonated Lys355 (top) or deprotonated Lys355 (bottom), are shown. The geometric centres of the oxalate ion and the binding site residues were used to calculate the distance. Different colours represent the different oxalate ions included in the simulation. **d–f** MD simulations started from the oxalate-bound occluded OxlT crystal structure (PDB ID 8HPK). **d** The RMSDs from the initial occluded crystal structure are shown for three independent trajectories in different colours. **e** Hydrophobic gates of OxlT. In top panel, the numbers of water molecules within 15, 8, and 4 Å distance from the bound oxalate ion are plotted in brown, yellow, and red, respectively. In bottom panel, a snapshot at 1000 ns is shown in the zoom-out and close-up views. Water molecules are coloured as same as panel **b**. See also Supplementary Movie 1. **f** The observed transition from the occluded to the outward-facing conformation triggered by the Gln34 flip. The oxalate ion and binding-site residues are shown in the stick representation. Gln34 is highlighted with the red circle. Water molecules are shown in the vdW representation. Water molecules are coloured as same as panel **b**. The broken lines between the Thr38 side-chain and the Val240 main-chain in black and red depict the distances those within or out of H-bonding, respectively. See also Supplementary Movie 2.

unaccompanied with the conformational transition in both the oxalate- and formate-bound forms (Supplementary Figs. 11 and 12). These findings suggest that the Gln34 side chain, in conjunction with the hydrogen bond between Thr38 and Val240, functions as a "latch of the periplasmic gate" to prevent the transition from the occluded to the outward-facing conformations. Indeed, the Q34A mutant displayed a partial loss of the binding and transport activities relative to the wild-type protein (Fig. 2c–e), indicating that the mutation destabilises of the occluded conformation. Gln34, along with Thr38, is strictly conserved within the OFA family (Supplementary Fig. 3).

The O-C-C-O dihedral angle of the oxalate ion in the occluded binding site became ~90° after the Gln34 flip (Supplementary Fig. 13a), which is the value observed in solution[43]. This contrasts with the other two trajectories without the Gln34 flip, where the oxalate dihedral angle remained around 40–50° (and its inverted position at 130–140°; Supplementary Fig. 13a), which is similar to those found in the crystal structure and the QM/MM calculations. Intriguingly, the values observed in the bound oxalate to the outward-facing OxlT during the simulation were broadly distributed with double peaks at ~60° and ~120° (Supplementary Fig. 13b), which differ from those in solution and rather closer to those in the occluded crystal structure. These findings suggest that the bound oxalate rearranges its conformation in response to the environmental change caused by OxlT conformational switching and adopts a favourable conformation for the subsequent step in the transporter cycle.

No opening of the cytoplasmic gate was observed during the 1 μs simulation for any of the trajectories from the occluded conformation. This may be attributed to the extensive interdomain interactions observed at the cytoplasmic side, such as the motif A involving charge relay networks (Fig. 2g) known to stabilise the outward-facing conformation[38]. These results suggest that the transition from the occluded to the inward-open state has a slow kinetics among the entire transport process. One of the enigmatic mutants showing reduced oxalate binding but retained transport activity, Y124A (Fig. 2c–e), locates at the entrance of the cytoplasmic gate underneath the bound oxalate (Figs. 2f and 4f). The mutation might destabilise the hydrophobic layer at the gate and facilitate its opening, probably the rate-limiting step of the transport, and thus could compensate for the reduced affinity to oxalate in its transport activity.

## Discussion

The two crystal structures of OxlT and the MD simulations based on them provided clues to understand the alternating access transport process of OxlT (Fig. 5). The following process is described according to the electrochemical gradient formed in *O. formigenes* within the gut. For the oxalate uptake process, OxlT exhibits an extensive positively charged surface in the cavity open to the periplasm, allowing a binding of acidic oxalate to the binding site. The positively charged surface also avoids the conformational transition to the next transport step in the absence of the substrate that is an indispensable characteristic for an antiporter. Nevertheless, the oxalate binding neutralises the local

positive charge and enables the conformational switch from the outward-facing conformation to the occluded conformation. Furthermore, the calculated relative binding free energies of oxalate to OxlT revealed a significant stabilisation in the occluded conformation compared to the outward-open conformation (i.e., ~20 kcal/mol decrease), which provides a physical basis for the conformational switch induced by oxalate binding (see "Methods" section and Supplementary Table 4). The occluded state is an essential step for transport to serve as a discriminatory checkpoint between oxalate and necessary host metabolic intermediates, such as those in the Krebs cycle, using the size restriction imposed by the binding pocket. The occluded conformation may eventually allow opening of the cytoplasmic gate and release of oxalate to the cytoplasm.

Subsequently, a formate binding to the inward-facing OxlT may return the transporter to the occluded state. The conformational transition required for returning from the occluded to the initial outward-facing states in the antiport cycle can be achieved by a transient flip of a side chain of a substrate-neighbouring residue, Gln34, and disruption of the hydrogen bond between Thr38 and Val240. The conformational landscape plotted by the periplasmic gate (Thr38–Val240) and cytoplasmic gate (Met128–Pro332) distances[42,44] sampled in the MD simulations shows that the order parameters separate the occluded and outward-facing conformations well (red and yellow plots, Fig. 5 inset). Nevertheless, the only trajectory that accompanies the Gln34 flip shows a full transition covering the end-point occluded and outward-facing crystal structures (blue points in Fig. 5 inset). The mechanism of the conformational change that can be triggered by a single sidechain flip, along with the structural flexibility facilitated by the high glycine ratio, likely accounts for the rapid catalytic turnover exhibited by OxlT[12].

These structural observations imply that OxlT utilises the MFS architecture and evolved in accordance with favourable symbiosis between the host animals and gut microbes. The structural and functional characteristics of OxlT also likely underlie those of the other OFA family members. Approximately 2,000 OFA members are registered in the database[45], and all but OxlT are functionally uncharacterised. Therefore, this study also contributes to understanding unknown 'dark' protein families. Clarifying the inward-facing conformations of OxlT (a dotted circle in Fig. 5 inset) is the next challenge understanding the structural biology of OxlT.

## Methods
### Ethics statement
All animal experiments conformed to the guidelines of the Guide for the Care and Use of Laboratory Animals of Japan and were approved by the Animal Experimentation Committee at the University of Tokyo (permission number: RAC07101).

### Preparation of OxlT
C-terminal nona-His-tagged OxlT from *O. formigenes* strain OxB[46] was expressed in *E. coli* XL3 at 20 °C for 24 h with 1 mM isopropyl-β-

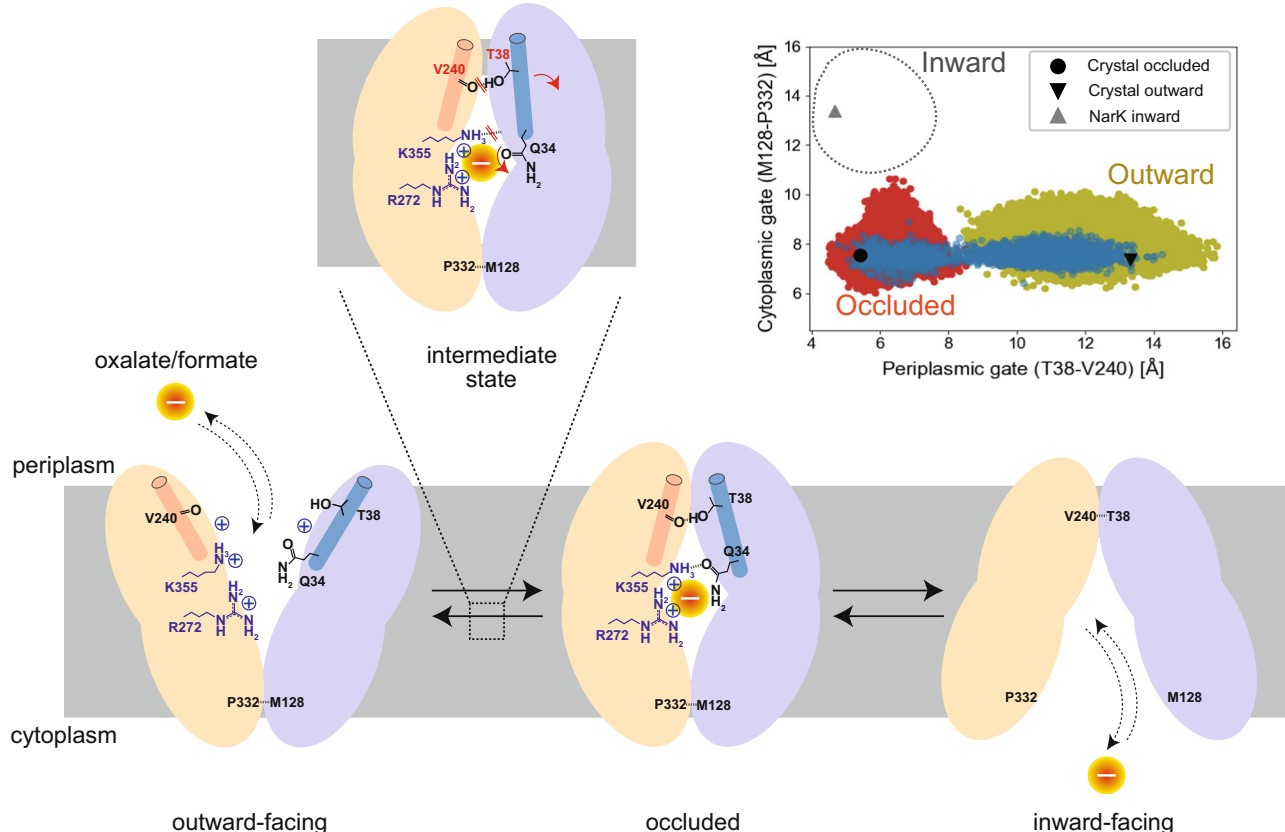

**Fig. 5 | Schematic drawing of the transport process and conformational switching of OxlT.** The conformational landscape of OxlT along the periplasmic and cytoplasmic gate distances is shown in the top right panel. The Cα distances of the gate residues in MD simulations of the occluded and outward-open states and Gln34-induced transition are shown in red, yellow and blue, respectively. The gate-residue distances in the current occluded (PDB ID 8HPK) and outward-facing (PDB ID 8HPJ) crystal structures as well as the Nark inward-facing crystal structure (PDB ID 4U4T) are also shown.

D-thiogalactopyranoside (IPTG)[47]. Bacterial cells were suspended in lysis buffer (50 mM Tris-HCl, 200 mM potassium acetate, 1 mM EDTA, 1 mM PMSF, 5 mM MgCl₂, 20 μg/mL DNaseI and 0.23 mg/mL lysozyme, pH 8.0) and then disrupted using EmulsiFlex C-5 (Avestin). Cell debris was removed by centrifugation (9,600×g for 30 min), and cell membranes were then collected by centrifugation (185,000×g for 1 h). The membrane fraction was solubilised with 40 mM n-dodecly-β-D-maltoside (DDM) in buffer A (20 mM HEPES-KOH, 200 mM potassium acetate, 10 mM potassium oxalate and 20% glycerol, pH 8.0) and applied to Ni-NTA Superflow resin (QIAGEN) or HisTrap FF crude (GE Healthcare) in an XK16 column (GE Healthcare). The column was washed with buffer A containing 1 mM DDM and 30–50 mM imidazole, and then protein was eluted with buffer A containing 1 mM DDM and 250 mM imidazole.

### Preparation of antibody fragments

A proteoliposome antigen was prepared by reconstituting purified functional OxlT at high density into phospholipid vesicles consisting of a 10:1 mixture of egg phosphatidylcholine (PC) (Avanti Polar Lipids) and adjuvant lipid A (Sigma) to facilitate an immune response. BALB/c mice (7-week-old females) were immunised with the proteoliposome antigen using three injections at two-week intervals. Mice were kept under a condition with a 12 h light/dark cycle at an ambient temperature of 23 ± 3 °C and humidity of 40–60%.

**D5901Fab.** Mouse monoclonal antibodies against OxlT were selected as previously described[48]. Antibody-producing hybridoma cell lines were generated using a conventional fusion protocol. Hybridoma clones producing antibodies that recognised conformational epitopes

in OxlT were selected by a liposome enzyme-linked immunosorbent assay (ELISA) on immobilised phospholipid vesicles containing purified OxlT, allowing positive selection of the antibodies that recognised the native conformation of OxlT. Additional screening for reduced antibody binding to SDS-denatured OxlT was used for negative selection against linear epitope-recognising antibodies. Stable complex formation between OxlT and each antibody clone was checked with fluorescence-detection size-exclusion chromatography. Whole IgG molecules, collected from the large-scale culture supernatant of monoclonal hybridomas and purified using protein G affinity chromatography were digested with papain, and Fab fragments were isolated using HiLoad 16/600 Superdex200 gel filtration followed by protein A affinity chromatography. The sequence of the Fab was determined via standard 5′-RACE using total RNA isolated from hybridoma cells.

**20D033Fv.** Single-chain Fv (scFv) fragments against OxlT were screened out from an immunised mouse phage displayed antibody library[49]. Immunised mice were euthanised, and their splenocyte RNA isolated and converted into cDNA via reverse-transcription PCR. The $V_L$ and $V_H$ repertoire was assembled via an 18-amino acid flexible linker and cloned into the phage-display vector pComb3XSS. Biotinylated proteoliposomes were prepared by reconstituting OxlT with a mixture of egg PC and 1,2-dipalmitoyl-sn-glycero-3-phosphoethanolamine-N-(cap biotinyl) (16:0 biotinyl Cap-PE; Avanti), and used as binding targets for scFv-phage selection. Targets were immobilised onto streptavidin-coated paramagnetic beads (Dynabeads) or streptavidin-coated microplates (Nunc). After four rounds of biopanning, liposome ELISAs were performed on periplasmic extracts of individual colonies.

Positive clones were collected and evaluated using a Biacore T100 (GE Healthcare).

Antibody scFv fragments are undesirable for use as crystallisation chaperones because they can intermolecularly form domain-swapped dimers, and the dimer-monomer equilibrium may increase structural heterogeneity. Therefore, we used Fv fragments for crystallisation trials. The Fv fragment were expressed in *Brevibacillus choshinensis* using the iRAT system[50]. Culture supernatant was adjusted to 60% ammonium sulphate saturation, and the precipitate was pelleted, dissolved in TBS buffer (10 mM Tris-HCl, pH 7.5, 150 mM and NaCl) and dialysed overnight against the same buffer. Dialysed proteins were mixed with Ni-NTA resin equilibrated with buffer B (10 mM Tris-HCl, pH 7.5, 150 mM NaCl and 20 mM imidazole). Bound proteins were eluted with buffer C (10 mM Tris-HCl, pH 7.5, 150 mM NaCl and 250 mM imidazole), mixed with TEV-His$_6$ and dialysed overnight against TBS buffer. Cleaved His$_6$ tag and TEV-His$_6$ were removed using a HisTrap column equilibrated with buffer B. The tag-free Fv fragment was concentrated and loaded onto a HiLoad16/60 Superdex75 column (GE Healthcare) equilibrated with TBS buffer. Peak fractions were pooled, concentrated, flash frozen in liquid nitrogen and stored at −80 °C.

## Crystallisation

For crystallisation of oxalate-bound OxlT complexed with D5901Fab, the purified OxlT was mixed with purified D5901Fab at a 1:1.3 molar ratio at 4 °C overnight and applied to a HiLoad 16/60 Superdex200 pg column (GE healthcare) using buffer D (20 mM MES-KOH, 200 mM potassium acetate, 10 mM potassium oxalate, 20% glycerol, and 0.51 mM DDM, pH 6.2) as running buffer. Purified sample was dialysed in buffer E (20 mM MES-KOH, 10 mM potassium oxalate and 0.51 mM DDM, pH 6.2). Crystals were obtained by the sitting-drop vapour diffusion method at 20 °C by mixing purified sample (-10 mg/mL) with a reservoir solution of 0.1 M sodium citrate, pH 5.5, 0.05 M NaCl and 26% (v/v) PEG400. Crystals were frozen in liquid nitrogen in advance of data collection.

For crystallisation of ligand-free OxlT complexed with 20D033Fv, purified OxlT was mixed with purified 20D033Fv at a 1:2 molar ratio at 4 °C overnight and purified using Superdex200 Increase 10/300 GL (GE healthcare) in buffer F (20 mM MES-KOH, 100 mM potassium acetate, 10 mM potassium oxalate and 0.02% DDM, pH 6.2). Purified sample was reconstituted into a lipidic mesophase. The protein-LCP mixture contained 50% (w/w) protein solution, 45% (w/w) monoolein (Sigma) and 5% (w/w) cholesterol (Sigma). The resulting lipidic mesophase was dispensed as 50 μL drops into 96-well glass plates and overlaid with 0.8 μL of precipitant solution using an NT8-LCP crystallisation robot (Formulatrix) and were then covered with thin cover glasses. Crystallisation setups and the 96-well glass sandwich plates (Molecular Dimension) were incubated at 20 °C. Crystals were obtained in a week under the following precipitation conditions: 100 mM Glycine, pH 9.0, 26–36% (v/v) PEG400, and 50–150 mM MnCl$_2$. Crystals were harvested directly from the lipidic mesophase using Mesh Litholoops (Protein Wave) and flash cooled in liquid nitrogen.

## Data collection and structure determination

X-ray diffraction data for oxalate-bound OxlT and for ligand-free OxlT were collected at 1.0 Å at the SPring-8 beamline BL41XU using MX225HE (Raynoix) and BL32XU using an EIGER X 9 M detector (Dectris, Ltd), respectively, under a control of BSS[51] and a cryostream operating at 100 K. Data were merged, integrated and scaled to 2.6 Å (oxalate-bound OxlT) and 3.1 Å (ligand-free OxlT) using the KAMO system[52], which exploits BLEND[53], XDS[54] and XSCALE[55] (Supplementary Table 1). Data were corrected for anisotropy using the STARANISO server[56]. The correction deleted many weak reflections with very low spherical completeness in the higher resolution shells. For refinement,

we used data to 3.0 Å (oxalate-bound OxlT) and 3.3 Å (ligand-free OxlT) that contained more than 25% (oxalate-bound OxlT) and 22% (ligand-free OxlT), respectively, of the data in the highest shell. The crystal structure was solved using molecular replacement with PHASER[57]. The search models were structures of N- and C-terminal halves of the glycerol-3-phosphate transporter GlpT (PDB ID: 1PW4)[58] and an Fab fragment (PDB ID: 1XF4)[59] for oxalate-bound OxlT, and structures of N- and C-terminal halves of oxalate-bound OxlT determined in this study (residues 11–199 and 204–404, respectively) and a scFv fragment (PDB ID: 5B3N)[60] for ligand-free OxlT. Structure models were manually rebuilt with COOT[61] and refined with Phenix[62]. In the ligand-free OxlT crystal, two units of OxlT (chain A and D) were found in an asymmetric unit. No significant structural difference was observed between the two (Cα RMSD of 0.365 Å for residues 15–410). Data collection and refinement statistics are shown in Supplementary Table 1 and 2. Ramanchandran statistics analysed with MolProbity[63] were 97.8% favoured, 2.2% allowed and 0.0% outliers for oxalate-bound OxlT, and 97.5% favoured, 2.5% allowed and 0.0% outliers for ligand-free OxlT.

## Ligand-binding assay

Ligand-binding of OxlT was evaluated using the GFP thermal shift assay (GFP-TS)[35]. The expression vector for the C-terminal GFPuv-fusion OxlT was constructed previously[23], and mutations were introduced into the vector via PCR using PrimeSTARMax (Takara Bio) and the oligonucleotide primers listed in Supplementary Data 1. The protein was expressed in BL21(DE3) basically as described in the subsection "Preparation of OxlT," where the protein production was started at an OD$_{600}$ of 0.8–0.9 and induced at 20 °C for 20 h.

Bacterial cells expressing the C-terminal GFPuv-fusion OxlT were suspended in buffer G (20 mM HEPES-KOH, 300 mM potassium acetate, pH 7.0) and disrupted by sonication using UD201 (TOMY). To obtain a membrane fraction, cell debris was removed by centrifugation (17,800×$g$ for 15 min at 4 °C), and the supernatants were then centrifuged at 252,000×$g$ for 30 min at 4 °C. The membrane fractions were washed twice by repeating the suspension with buffer G followed by ultracentrifugation at 252,000×$g$ for 30 min at 4 °C. Membrane fractions from a 20 mL culture were solubilised with 240 or 480 μL of buffer G containing 1% (w/v) DDM. After a solubilisation at 4 °C for 1 h, the insoluble materials were removed by ultracentrifugation 161,000×$g$ for 20 min at 4 °C.

For mutant assays, the detergent-solubilised wild-type and mutant OxlT-GFPuv proteins were diluted with buffer G containing 1% DDM to adjust the OxlT-GFPuv concentration giving the same fluorescent intensities. The solutions were then incubated for 1 h on ice in the presence or absence of 3 mM potassium oxalate. For ligand assays, the wild-type OxlT-GFPuv solution was incubated in the presence or absence of 10 mM sodium oxalate, sodium malonate or sodium succinate. After incubation, octyl-β-D-glucoside in buffer G was added to a final concentration of 1%, and the solutions were subjected to heat denaturation at 30–80 °C for 10 min using a C-1000 Touch Thermal Cycler (BIO-RAD). The aggregated fractions were then removed by centrifuging the solutions at 10,740–15,340×$g$ (14,000 rpm using a TOMY PCR96-02 rotor, depending on the tube positions in the rotor) for 20 min at 4 °C. The resultant supernatants were aliquoted into a 384-well Low Volume Black Microplate (Corning), and the fluorescence intensities were measured using a Varioskan Flash (Thermo Fisher Scientific) at an excitation wavelength of 395 nm and an emission wavelength of 507 nm. The observed fluorescence intensities were subtracted with that of the buffer background and normalised with setting the values of the sample without heat denaturation as 1 and the background as 0. The apparent melting temperature ($T_m$) values were estimated by fitting the fluorescence intensity values to the Gibbs–Helmholtz equation[64] using Kaleidagraph (Hulinks), assuming that the protein of interest is in an equilibrium between a folding and an unfolding state and setting the enthalpy and heat capacity changes

of unfolding ($\Delta H$, $\Delta C_p$) and $T_m$ as independent variables. The data were analysed by two-sided one-way analysis of variance with Dunnett's test using the wild-type (WT) as a reference in Prism 8 (GraphPad). The statistical significance is defined as *$P < 0.05$.

## Transport assay

Transport activity of OxlT was evaluated using both an *in cellulo* system with recombinant *E. coli* cells and an in vitro system with proteoliposomes reconstituted with the purified OxlT. For mutant OxlT assays, mutations were introduced into the OxlT expression vector, pRSF-OxlT[36], via PCR using PrimeSTARMax and the oligonucleotide primers listed in Supplementary Data 1.

In the *in cellulo* assay, OxlT activity was measured by coupling with light-driven inward proton transfer by xenorhodopsin co-expressed in *E. coli*, as previously described[36]. Briefly, *E. coli* BL21 (DE3) cells transformed with the expression vectors for OxlT and xenorhodopsin were cultured at 37 °C, and protein production was induced by the addition of 1 mM IPTG and 10 μM all-*trans* retinal (Sigma) at an absorbance of 0.8–0.9 at 600 nm. After a culture at 20 °C for 20 h, *E. coli* cells were collected by centrifugation and suspended with 50 mM $K_2SO_4$ to a cell density with OD at 660 nm of ~10. The light-induced pH change of the cell suspension was monitored with a pH electrode at 25 °C using continuous stirring. The cell suspension was first placed in the dark until the pH of the sample stabilised. The sample was then illuminated using a Xe lamp through a Sharp Cut Filter Y44 (a longpass filter at ≥420 nm, HOYA) for 10 min, and the pH change in the absence of oxalate ($\Delta pH_O$) was monitored. The light intensity was adjusted to ~150 mW/cm$^2$ at 550 nm using an optical power metre and an optical sensor. The illuminated sample was placed back in the dark and when the pH stabilised, 5 mM potassium oxalate was added to the sample to enable transport via OxlT for 10 min. The sample was again illuminated under the same condition as above, and the pH change ($\Delta pH_S$) was monitored. The transport activity was evaluated by the difference in pH change ($\Delta\Delta pH$) between $\Delta pH_S$ and $\Delta pH_O$; this was further corrected by subtracting the background differential pH change ($\Delta\Delta pH$) measured with *E. coli* expressing xenorhodopsin alone. The activities for each mutant were normalised by the corrected $\Delta\Delta pH$ and the relative expression level, analysed by western-blotting using Penta•His Antibody (QIAGEN, Cat. No. 34460, 1:2000 dilution), of wild-type OxlT measured on the same day of experiment (Supplementary Fig. 14; the uncropped blots are in the Source Data file). We also performed an assay for the Y150A mutant; however, this mutant affected the expression level of xenorhodopsin due to unknown reasons (Supplementary Fig. 14), and we therefore excluded the Y150A result from this paper.

In the in vitro assay, the WT and mutant OxlT proteins were expressed and purified as described above and in the "Preparation of OxlT" subsection with minor modifications. The OxlT proteins were eluted from Ni-NTA resin in the absence of glycerol in the elution buffer and purified on a size-exclusion chromatography column Superdex 200 10/300 GL (GE Healthcare) equilibrated with 20 mM MES-KOH, 200 mM potassium acetate, 0.2 mM DDM, pH 6.2. The monomeric fractions were pooled, concentrated to ~1 mg/ml, supplemented with 20% (v/v) glycerol, quantified, aliquoted, and flash-frozen for further use. Proteoliposomes were prepared similarly to that described in Lee et al.[65]. *E. coli* total lipids extract (Avanti, in chloroform) was aliquoted into glass tubes, and the solvent was evaporated for more than 5 min at room temperature under a stream of nitrogen gas. In order to obtain unilamellar vesicles, dried *E. coli* lipids were resuspended at 6.7 mg/ml in a solution containing 50 mM MOPS-KOH and 10 mM potassium formate, pH 7.0, and sonicated for 45 sec in a water-bath sonicator. The unilamellar vesicle was reconstituted with the purified OxlT variants at 200:1 (w/w) lipid:protein ratio or with the equivalent volume of the control buffer (50 mM MOPS-KOH with 1.57 mM DDM, the same volume as the OxlT solution) in microtubes by

three freeze-and-thaw cycles using liquid nitrogen and water baths. Reconstituted unilamellar vesicles were sonicated three times for 15 s in total using a hand-probe device on ice to form (proteo)liposomes with a diameter of <200 nm.

Oxalate uptake assays were initiated by adding 50 mM potassium oxalate to the liposome solution at 20 °C. Before starting the experiments, the AG1-X8 anion-exchange resin (BioRad) was equilibrated with a 1:1 (w/v) volume of 150 mM sodium acetate, pH 8.2, and prepared by centrifuging 400 μL slurry for 1 min at 700×*g* and 4 °C on a spin column. In all, 50 μL reaction was collected either immediately or after incubation for the time indicated in the figures. By applying the reaction solutions onto the AG1-X8 spin columns and centrifuging for 1 min at 700×*g* and 4 °C, the oxalate ions outside the liposomes were removed, and the samples were collected in a microtube. Subsequently, 5 μL of a 10% (w/v) Triton-X100 (Nacalai tesque) solution was added to each sample to dissolve liposomes containing oxalate ions that were either transported into or adhered to the liposome. The relative oxalate concentrations were determined utilising the Oxalate Oxidase (OxOx) Assay Kit (Abcam) coupled with an oxidation reaction using recombinant OxOx (*B. subtilis*, Biovision). 10 μL liposome lysate was mixed with 10 μL detection buffer (9 μL OxOx Assay Buffer, 0.4 μL OxOx Converter, 0.2 μL Red Probe, 0.02 μg OxOx, up to 10 μL with water) in each well of a low volume 384-well plate (Corning) with a multichannel pipette. The plate was immediately read by Varioskan Flash in fluorometric kinetics mode with excitation at 535 nm (12 nm bandwidth), emission at 587 nm, the integration time of 200 msec, and 20-sec intervals for a total of 83 min at room temperature. The initial slopes of the fluorescence–time curves were determined by linear regression on the first 600-s data and interpreted as the initial velocities of the OxOx reaction, which were linearly correlated with the oxalate concentration under this assay condition (Supplementary Fig. 6). At least two independent liposome preparations and oxalate transport assays were performed for the OxlT variants and negative control, and OxOx assays were conducted in duplicate or quadruplicate for each condition. The final bar graph displays the normalised values to the mean of the WT data at 60 min.

The data (those at 60 min in the case for the in vitro assay) were analysed by a two-sided one-way analysis of variance with Dunnett's test in Prism 8 using the WT as the reference. The statistical significance is defined as *$P < 0.05$.

## Molecular dynamics simulation

The OxlT crystal structures were used as initial structures, with missing residues at the central loop modelled with MODELLER[66]. Protonation states were analysed using PROPKA 3.1[67,68], with the default parameter. Based on the analysis, Lys355 exhibits a deviated p$K_a$ value of 7.00 in the outward-facing structure. This deviation was not observed in the occluded structure (p$K_a$ value of 8.61). Thus, both protonation states for Lys355 were considered in the outward-facing state. The OxlT protein was embedded in the membrane using the Membrane Builder plugin in CHARMM-GUI[69,70]. A 1-palmitoyl-2-oleoylphosphatidylethanolamine (POPE) bilayer with a length of 120 Å for the *x* and *y* dimensions was used. The PE lipid is a major component in both *O. formigenes*[71], from which OxlT is derived, and *E. coli*[72], in which transport assays were conducted. In addition, there is no evidence that other specific lipids are required for OxlT activity. The protein–membrane system was solvated with TIP3P water molecules and 150 mM KCl, resulting in the *z* dimension length of 100 Å. Then, all 87 Cl⁻ were replaced with 58 oxalate ions using AmberTools17[73] without altering the total charge by taking into account the scaled effective charge (−1.5e) of the oxalate model in solution (see ECCR below). The final MD system contained 146015 and 143611 atoms for the occluded and outward-facing OxlT system, respectively. MD simulations were then performed using NAMD 2.12[74]. The Amber ff14SB and Lipid14 forcefields were employed to describe the protein and the membrane, respectively[75,76]. The oxalate ligand in

solution was described with parameters determined by the electronic continuum correction with rescaling (ECCR), based on Ab Initio Molecular Dynamic simulation, developed by Kroutil et al.[43,77]. The oxalate ligand in the binding site of OxlT was described using parameters determined by the Restrained Electrostatic Potential (RESP) scheme[78] without applying the ECCR correction, considering that the protein environment differs from that of water solution. The RESP charges have been calculated by the Antechamber software[79]. To our knowledge, no study has yet simulated oxalate complexed with protein, although several MD studies have simulated the solvation of oxalate anion in bulk, its complexation with calcium cation, and the adsorption surface process[80–82]. The MD system was set up with a minimisation for 10,000 steps, heated from 0 to 10 K with a step of 0.1 ns per degree in NVT ensemble, then 10 to 310 K in NPT with a step of 0.2 ns per 30 degree, and 10 ns of equilibration with NPT ensemble simulation at 310 K. Then, production runs of 1.0 and 1.7 μs in NPT conditions were performed for the occluded and outward-facing OxlT (for each protonation state of Lys355) system, respectively. A temperature of 310 K was maintained with the Langevin thermostat, with the pressure set to 1 atmosphere using the Nosé-Hoover Langevin piston. Periodic boundary conditions were applied, and long-range electrostatic interactions were treated by the particle mesh Ewald method with a real space cut-off of 12 Å and a switch function at 10 Å. The integration time step was 2 fs.

To establish the simulation system with formate, a carboxylate moiety of the oxalate pointing toward to the Lys355 was replaced with a hydrogen atom in the oxalate-bound occluded structure to generate the initial structure of the formate-OxlT complex. In addition, a K$^+$ ion was removed from the prior model and replaced with a water molecule to account for the loss of a negative charge. GAFF force field parameters[79] were used for formate. The same equilibration and production protocols as described above were performed. The full relaxation of the OxlT protein at the end of the equilibration step guarantees a good adjustment of the binding site for a smaller ligand as well as a realistic conformation for production runs.

As a summary of the simulation systems, we have constructed four systems with different combinations of the initial structure (outward-open or occluded structure), the protonation state of Lys355, and the bound ligand (oxalate or formate; in the case for the occluded structure). In Supplementary Table 5, these simulation systems, as well as the number of trajectories and total simulation time, are summarised.

The relative binding free energy of oxalate to OxlT was calculated by the MM/GBSA method implemented in the programme MMPBSA.py[83]. In the MM/GBSA method, the binding free energy is decomposed into gas-phase and solvation energies, which are calculated by the molecular mechanics force field (MM) and the generalised Born implicit solvent model with the solvent accessible surface area (GBSA), respectively. Note that the entropy effect was not included in the calculation. We analysed a trajectory depicting the conformational transition from the occluded to the outward-open state (Fig. 4d, f and Supplementary Fig. 13a). The trajectory was divided into three stages: the occluded OxlT with the oxalate dihedral ~50° (0–40 ns; denoted as OxlT-occ-dih50), the occluded OxlT with the oxalate dihedral ~90° (41–320 ns; OxlT-occ-dih90) and the outward-open OxlT with the oxalate dihedral ~90° (321–1000 ns; OxlT-op-dih90). The MM/GBSA binding free energy was determined for each trajectory stage (see Supplementary Table 4).

The water density during the simulation was calculated by a module from MDAnalysis[84] after the protein was centred and superimposed.

## QM/MM calculation
Several QM/MM models were employed with the oxalate-bound structure (PDB ID 8HPK) to assess the relevance of the binding site environment for the internal conformation of oxalate. First, the oxalate ligand was assigned to the QM part while the whole protein was

assigned to the MM part. Second, the first shell of residues that interact directly with the ligand, (Gln34, Tyr35, Tyr124, Arg272 and Lys355) were added to the QM part. Third, the second shell of the binding site (Tyr150, Trp324, Tyr328 and Trp352) were added to the QM part to build a full binding site environment surrounding the oxalate ligand. All the QM/MM calculations were performed with ONIOM[85], implemented in Gaussian 16[86]. The density functional theory (DFT) method[87,88] was used to treat the QM region at the B3LYP/6-31 + G(d,p) level of theory[89,90], including Grimme's dispersion correction with Becke–Johnson damping (D3BJ)[91]. The MM region of the system was described by the same force field as that in the MD simulations. The electronic embedding scheme was used such that the MM region polarises the QM electronic density. An explicit link atom was added between the α and β carbons for each residue located in the QM region to handle the covalent boundary between the QM and MM parts. Minima of the potential energy surface were confirmed by having no imaginary frequencies. Additional pure DFT calculation of oxalate ligand with fixed side chains of the binding site residues were performed with the same QM level of theory as in the QM/MM calculations. As with QM/MM calculations, optimised structures, obtained as stationary points on the potential energy surface, were true energetical minima without imaginary frequencies.

## Reporting summary
Further information on research design is available in the Nature Portfolio Reporting Summary linked to this article.

## Data availability
Coordinates and structure factors for OxlT have been deposited in the Protein Data Bank under the accession numbers 8HPK (OxlT-fab complex; oxalate-bound occluded form)[92] and 8HPJ (OxlT-Fv complex; ligand-free outward-facing form)[93]. The MD-related data have been deposited in the Zenodo repository [https://doi.org/10.5281/zenodo.7597686][94]. Coordinates with the PDB IDs 1PW4[58], 1XF4[59], 5B3N[60], 4U4W[95], 4U4T[96] and amino acid sequences of OFA family (IPR026355) were used in this study. Source data are provided with this paper.

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

## Acknowledgements

We thank Drs. Kazuya Hasegawa, Hideo Okumura, Yoshiaki Kawano and Kunio Hirata, SPring-8, for the X-ray diffraction data collection support; Yayoi Nomura, Yoshiko Nakada-Nakura and Yumi Sato for their technical assistance in the generation of antibodies; Kyoko Fujita for their technical assistance with the functional assays; and Professor Keietsu Abe for their valuable advice on the OxlT functional assays. The synchrotron radiation experiments were performed at the BL41XU and BL32XU of SPring-8 and, with approvals of the Japan Synchrotron Radiation Research Institute (JASRI) (Proposal No. 2012B1096, 2015A1080,

2015B2080). Computations were partially performed using Research Center for Computational Science, Okazaki, Japan (Project: 21-IMS-C175, 22-IMS-C189). This work was financially supported by JSPS KAKENHI Grant Numbers JP20H03195 (to A.Y.), JP18H02415 (to K.O.), JP26440086 (to T.H.) and Research Fund from Koyanagi Foundation (to A.Y.), Takeda Science Foundation (to Ta.S.), the Platform Project for Supporting Drug Discovery and Life Science Research [Basis for Supporting Innovative Drug Discovery and Life Science Research (BINDS)] from AMED under grant no. JP20am0101079 (to S.I.). The authors would like to thank Enago (www.enago.jp) for the English language review.

## Author contributions

Te.H. and A.Y. conceived the study. Te.S., M.Y., Ta.S., K.H. and N.N. performed protein purification. N.N., H.I., Ta.H., and S.I. performed antibody preparation. Te.S., M.Y., A.Y, Te.H., Ta.S., and K.H. performed crystallization and X-ray data collection. Ta.S., Te.H., M.H., and A.Y. performed the structure analysis. M.H., N.T., Yut.S., K.K., A.Y., and Yuk.S. performed the functional assays. T.J.L. and K.O. performed molecular dynamics and QM/MM simulations. T.T. performed a preliminary molecular dynamics simulation. A.Y., K.O., Ta.S., N.N., T.J.L., M.H., N.T., and Yut.S. wrote the paper, together with input from all of the other authors.

## Competing interests

The authors declare no competing interests.
