## [Peer Review File · Nature Communications]

REVIEWER COMMENTS

Reviewer #1 (Remarks to the Author):

In this manuscript titled “Structure and mechanism of oxalate transporter Oxlate T in an oxalate-degrading bacterium in the gut microbiota” submitted by Atsuko Yamashita, et al, the authors determined two high-resolution crystal structures of an oxalate transporter OxIT in its apo and an oxalate-bound conformations, solved a long-standing problem. OxIT was extensively studied 20 years ago. This major facilitator transporter catalyzes precursor-product exchange, which is linked to the generation of metabolic energy. It is an interesting system in bioenergetics and also plays a role in preventing kidney stone formation. The structures of OxIT at the apo and oxalate-bound conformations were refined to a resolution of 3.3 Å and 3.0 Å assisted with the aid of Fv or Fab fragments, respectively. The apo structure was trapped in an outward-open conformation, and the oxalate-bound structure was trapped in an occluded conformation. All-atom molecular dynamics simulations were used to simulate the binding interactions of oxalate to OxIT, which also provided critical insights into its coupled conformational changes. The structural information is consistent with the previous biochemical and functional studies. In general, this study seems technically sound, scientifically rigorous, and presented logically. This beautiful work could be further improved by addressing the following comments/suggestions.

Major comments:

1. On page 4, line 101, “We confirmed that the Fab fragment used for crystallisation binds to OxIT both in the presence and absence of oxalate,”. There is no data or citation to support this statement. Are there any functional binding data available to support that oxalate binds to the Fab/OxIT or Fv/OxIT complexes?
2. A brief introduction about the Fab and Fc productions will be helpful.
3. On page 15, line 339 -341, “the oxalate binding neutralises the local positive charge and enables the conformational switch from the outward-facing conformation to the occluded conformation”. The authors know clearly that the occluded state is essential for the transport function catalyzed by all MFS transporters (whether uniporter, symporter or antiporter). However, neutralization does not necessarily cause the conformation to move from outward-facing state to the occluded state. On the other hand, this research carried out detailed analysis of the configuration of the bound oxalate, it seems that the O-C-C-O dihedral angle is correlated to the conformation states well. I request the authors to calculate the oxalate binding free energy ΔG for both open and occluded states, and analyze the correlations between binding affinity, dihedral angle, and conformation.

If smaller dihedral angle in the occluded state is associated with significantly higher binding affinity, then an important mechanism can be hypothesized. Thus, for an antiporter, the substrate binding induced-fit to its high-affinity state triggers the conformational switch from open to closed states. In a symporter, it is likely this is achieved by cooperative binding as proposed in the Na/melibiose transporter.

4. With regard to the transport assay, a diagram and detailed explanation will help readers to understand this indirect assay. The western blot for all mutants and the co-expressed xenorhodopsin are needed to present.

On page 21, line 489, “the pH change”. Please define increase or decrease, and make this correction for all other places.

On page 21, line 476, “coupling with light-driven inward proton transfer”. Did you mean “the coupled light-driven inward proton transfer”? Can you detect format in the media?

The data presented in the figure 2C show that most mutants have significant transport activity. With this indirect measurement on pH change, instead of the substrate of OxIT, and also only single time-point data from a 10-min incubation, it is not known what is the true mutational effect on the oxalate transport by OxIT. What is coupling ratio between H⁺ flow and oxalate transport?

5. OxIT is a high-turnover number MFS transporter. Does the simple one-sidechain switching mechanism also contribute to the high-turnover number, in addition to the observed excess Gly residues in the protein?

Minor comments:

1) On page 16 line 367, add the OxIT source. Please clearly state from which bacterial strain the OxIT structure was made available.

2) On page 15, line 336, “allowing a spontaneous binding”. In fact, all binding events are spontaneous. I would delete “spontaneous” here and in other places.

3) Extended Data Table 2 needs an extensive table legend.

4) O1-C1-C2-O2. Please define oxalate atom position in the crystal structure.

5) Label the helical number in the figure 2 DEF

6) Figure 4, the stick size is too big. It is difficult to view the substrate.

7) In the Extended Data table 1 on the first row, add the important information (Apo and Oxalate-bound state, as well as their associated conformations). The Rmerge values are too high since including the unused poor data beyond 3.0Å resolution. The data quality that reflects the structure is not available. Please report the truncate the dataset to the refined levels.

8) Cite the pdb id in all figures and related text to distinguish the two crystal structures, and the crystal structure and MD simulations models.

Reviewer #2 (Remarks to the Author):

In this contribution by Jaunet-Lahary et al., the structure of an oxalate transporter in the presence and in the absence of a substrate is presented. The obtained resolution is much higher than previously reported (3 Å vs 6.5 Å) allowing assignment of side chains and importantly a substrate. I have no concerns about the structural work (apart from representation, see below), but my main concern is about the functional characterization of OxIT. The authors present it in a such minimal way that it looks suspicious. First of all it is absolutely indirect as they do not observe any transport of oxalate / formate, but monitor the acidification of the e.coli cells via inward proton pumping xenorhodopsin. The difference between the scenario (1) in the absence of oxalate, and the scenario (2) exposure to oxalate followed by illumination is not shown, and perhaps it is not even significant, and furthermore the internal buffer capacity of a cell is not taken into account and the impact of all other proton-transporting proteins is not considered. This experiment should be done preferably in vitro on OxIT reconstituted into proteoliposomes and in the best case with ¹⁴C-oxalate uptake assay for the direct visualisation of transport (which would also allow a proper characterisation of kinetics and not only speculation about it). In addition - affinity measurements are highly recommended, as so far there is only some qualitative indications (C2 binds stronger than C3 and C3 stronger than C4).

The authors put the great emphasis on the role of Q34, which they propose as a trigger to switch between open and occluded conformations. Although it seems feasible from their simulations, it is in a conflict with their functional data, where Q34A mutant retains at least 30% transport activity (that in principle should lead either to an arrest in a certain conformation, hence no transport, or deregulate the conformational change and lead to ~50% activity - should be possible to check with MD simulations on Q34A mutant).

The fact of abundant Gly residues in this protein is noteworthy - and again the role of these Gly for proposed flexibility required for transport can be and should be checked via mutagenesis (in vitro and in silico)

Minor issues:

line 117: 1.5~1.6 Å should be ~1.5-1.6 Å

line 124 and throughout the text: when you talk about certain residues it would be helpful to emphasise if they are conserved; these 52 glycines is actually the most likely reason why this protein is unstable without Fabs;

line 153 - it is not a substituting group, hence instead of carbamoyl should be carboxamide;

line 189 - supplanting? - replacement with?

line 499 - western blots are mentioned, but not shown;

Fig 1. Positions of Gly can be moved to Supplementary;

Fig 2C - no negative control is shown, no statistical information given; 2B - the numbers of helices perhaps are not really necessary;

Supplementary Fig 3A - the colour of OMIT map should be changed (red is usually used for negative peaks); panel C - show the additional H-bond (change view);

Supplementary Fig 6 -should be enlarged, hard to see.

To summarise - it is a good structural work with a rather poor functional characterization and it is more suitable for a more specialised journal.

Reviewer #3 (Remarks to the Author):

The manuscript reports on crystal structures (~3Å resolution) of oxalate-bound and ligand-free bacterial oxalate transporter, Ox1T, in two conformations, occluded and outward-facing. The authors analyze the structural data and perform molecular dynamics (MD) simulations to conclude that two basic residues (Lys355 and Arg272) form salt bridges with the substrate, the dicarboxylate oxalate. The MD simulations show that the flip of a single side chain (Lys355) neighboring the substrate is sufficient to trigger the opening of the protein to allow substrate passage. The authors' results further support the notion of Ox1T functioning as an antiporter between oxalate and formate, rather than a uniporter of each molecule.

The structural data from x-ray crystallography provides the community with further insight into the architecture of Ox1T, and into the mechanism by which oxalate is recognized as a C2 (rather than C4) substrate. The authors describe how the conformational change between the occluded and outward-

facing states is not simple a tilt of rigid helices but rather due to the flexible bending of transmembrane helices. A relatively high (12.4%) glycine content enables this plasticity.

The work is a contribution to the understanding of how Ox1T (and other transporters) can selectively bind substrates. That Ox1T functions as an antiporter is known, and this study shows atomic-level evidence of why this behavior is achieved.

Overall, the methodologies of crystallization, structure refinement, and simulation appear to be sound (standard), but the simulations cannot be reproduced as they are described in this manuscript (see comments below). The interpretation of the simulation results seems plausible, but the interpretation is not thoroughly supported by the reported results. The mechanism of substrate binding/unbinding, based on a single observed opening of a salt-bridge, is not rigorously proven or put into context of experimental evidence (kinetic assays). Additional evidence and experimental details are required.

The manuscript is clearly written but mixes present and past tense in many instances (see point 16 below). The discussion is nicely structured and easy to follow. The figures have been created with great care.

Below is a list of issues that must be clarified before being reviewed again for publication.

1) Could the authors please clearly list all models built, e.g. Model1=(oxalate+protein, Lys355 protonated), Model2=(formate+protein, Lys355 protonated), etc.... and make a table of all trajectories that were simulated corresponding to each model, including total simulation time. If several independent trajectories were run for a single model, please state in the table.

2) Please explain clearly, for each trajectory, what the starting geometry is, and whether it corresponds to one of the models built.

3) For each binding event observed, where is oxalate at t=0? Where is formate at t=0? Figure 4B suggests with the arrow that at t=0 oxalate is outside of the protein and enters the space between the NTD and CTD.

4) The authors write "several binding and unbinding events were observed for the neutral Lys355" or "Spontaneous binding of oxalate" was observed, but they never clearly define what a "binding" or "unbinding" event is. How is "binding" defined? What is a "single binding event"?

5) How long does the substrate occupy the “binding position”? In other words, what is the occupancy rate?

6) Figure 4C is of little help to understand binding. What exactly is being plotted as a function of time? Colors refer to different oxalate molecules, but how many independent trajectories are presented? How many oxalate molecules are present per model? What is the meaning of the distance signal >10Å not shown?

7) Is the flip of Lys355 side-chain due to charge repulsion with side chain of Arg272 a simulation event that can be reproduced? What happens if Lys355 is mutated to Ala? What happens if Arg272 is mutated to Ala?

8) What experimental kinetic assays (literature?) have been performed that may give us an idea of the reaction rates of oxalate binding/unbinding? Can the observed “binding event” be related to ΔG via transition state theory to experimental evidence?

9) The authors write that the binding event changes depending on the protonation state of Lys355. The authors also mention pH of the gut, a macroenvironment, but the pH of the gut is likely not the explanation for the deprotonation of a sidechain buried in a protein. Perhaps the authors should discuss how the protonation of Lys in buried hydrophobic pockets can change. Please check Takayama et al. “Direct Evidence for Deprotonation of a Lysine Side Chain Buried in the Hydrophobic Core of a Protein” in JACS 2008.

10) The authors seem to confuse atomic charges and simulation parameters in the discussion of QM and QM/MM. How are charges for the small molecules calculated? How are parameters (bonding and non-bonding) determined for these substrates? Have these charges and parameters been validated and tested? Are they robust? Are any other simulations involving oxalate or formate known? Given that the mechanism of binding relies heavily on the charges and parameters of the substrates, the quality of these computed numbers is significant for interpretation of results.

11) What is the “original RESP” scheme? What software was used?

12) Could the authors please comment on the choice of the membrane composition? Why was a homogenous membrane (phosphatidylethanolamine) chosen?

13) What are the x, y, z, dimensions of the final system? Only x,y dimensions of bilayer are given.

14) How many Cl⁻ ions are replaced in each model with oxalate ions? How is the model containing formate built if formate is constructed at one site (line 533)?

15) What is the integration time step (1 fs? 2 fs?) in the simulations?

16) Please stay in present tense when discussing results (e.g. line 111 should be “molecule displays”, line 112 should be “which is connected”, line 119 should be “are observed”, line 328 “results suggest”).

17) Lines 137-138 verb is missing, probably should read “is refined”?

18) What does “optimised” mean in this context? Structural refinement?

19) Lines 210-212 verb is missing

20) Line 278 should be “consisting of Thr38”

21) Lines 321-322 “change” and “changes” too close together

22) Line 532 should read “pointing toward Lys355” or similar

Responses to reviewers

Reviewer #1

In this manuscript titled “Structure and mechanism of oxalate transporter Oxlate T in an oxalate-degrading bacterium in the gut microbiota” submitted by Atsuko Yamashita, et al, the authors determined two high-resolution crystal structures of an oxalate transporter OxIT in its apo and an oxalate-bound conformations, solved a long-standing problem. OxIT was extensively studied 20 years ago. This major facilitator transporter catalyzes precursor-product exchange, which is linked to the generation of metabolic energy. It is an interesting system in bioenergetics and also plays a role in preventing kidney stone formation. The structures of OxIT at the apo and oxalate-bound conformations were refined to a resolution of 3.3 Å and 3.0 Å assisted with the aid of Fv or Fab fragments, respectively. The apo structure was trapped in an outward-open conformation, and the oxalate-bound structure was trapped in an occluded conformation. All-atom molecular dynamics simulations were used to simulate the binding interactions of oxalate to OxIT, which also provided critical insights into its coupled conformational changes. The structural information is consistent with the previous biochemical and functional studies. In general, this study seems technically sound, scientifically rigorous, and presented logically. This beautiful work could be further improved by addressing the following comments/suggestions.

Reply:

We appreciate the reviewer’s positive remarks regarding our study. We also value their insightful suggestions. Based on their feedback, we have revised our manuscript as follows.

Major comments:

1. On page 4, line 101, “We confirmed that the Fab fragment used for crystallisation binds to OxIT both in the presence and absence of oxalate,”. There is no data or citation to support this statement. Are there any functional binding data available to support that oxalate binds to the Fab/OxIT or Fv/OxIT complexes?

Reply:

We have confirmed that the antibody fragments used for crystallization bind to OxIT both in the presence and absence of oxalate. The results are now shown in Supplementary Fig. 1 (newly created).

2. A brief introduction about the Fab and Fc productions will be helpful.

Reply:

We added a brief introduction about the Fab and Fv production and further revised the first paragraph of the Results section as follows:

(L.97-106, p. 6)

“For the structural study of OxIT, an antibody-assisted crystallization strategy was adopted. In general, Fab or Fv antibody fragments that bind specifically to conformational epitopes in membrane proteins can increase the hydrophilic surface area available for the formation of rigid crystal lattices. Additionally, the bound antibody fragments can reduce the inherent protein flexibility and conformational heterogeneity, increasing the likelihood of successful crystallization of membrane proteins^{24,25}. OxIT was stabilised by binding two different antibody fragments, resulting in crystallisation under two different conditions. We confirmed that the antibody fragments used for crystallisation bind to OxIT both in the presence and absence of oxalate (Supplementary Fig. 1). Therefore, it is unlikely that the antibody fragments trap OxIT in a particular conformation artificially.”

3. On page 15, line 339 -341, “the oxalate binding neutralises the local positive charge and enables the conformational switch from the outward-facing conformation to the occluded conformation”. The authors know clearly that the occluded state is essential for the transport function catalyzed by all MFS transporters (whether uniporter, symporter or antiporter). However, neutralization does not necessarily cause the conformation to move from outward-facing state to the occluded state. On the other hand, this research carried out detailed analysis of the configuration of the bound oxalate, it seems that the O-C-C-O dihedral angle is correlated to the conformation states well. I request the authors to calculate the oxalate binding free energy ΔG for both open and occluded states, and analyze the correlations between binding affinity, dihedral angle, and conformation.

If smaller dihedral angle in the occluded state is associated with significantly higher binding affinity, then an important mechanism can be hypothesized. Thus, for an antiporter, the substrate binding induced-fit to its high-affinity state triggers the conformational switch from open to closed states. In a symporter, it is likely this is achieved by cooperative binding as proposed in the Na/melibiose transporter.

Reply:

We appreciate the reviewer’s thoughtful suggestion.

As suggested, we calculated the relative binding free energies of oxalate. The following sentences were added to the Method section.

(L.687-698, p.29-30)

“The relative binding free energy of oxalate to OxIT was calculated by the MM/GBSA method implemented in the program MMPBSA.py⁸¹. In the MM/GBSA method, the binding free energy is decomposed into gas-phase and solvation energies, which are calculated by the molecular mechanics force field (MM) and the generalized Born implicit solvent model with the solvent accessible surface area (GBSA), respectively. Note that the entropy effect was not included in the calculation. We analyzed a trajectory depicting the conformational transition from the occluded to the outward-open state (Fig. 4d and f; Supplementary Fig. 13a). The trajectory was divided into three stages: the occluded OxIT with the oxalate dihedral ~50 deg (0–40 ns; denoted as OxIT-occ-dih50), the occluded OxIT with the oxalate dihedral ~90 deg (41–320 ns; OxIT-occ-dih90), and the outward-open OxIT with the oxalate dihedral ~90 deg (321–1000 ns; OxIT-op-dih90). The MM/GBSA binding free energy was determined for each trajectory stage (see Supplementary Table 4).”

The calculated relative binding free energies demonstrated a ~20 kcal/mol decrease for OxIT-occ-dih50 compared to OxIT-op-dih90, indicating that oxalate binding is largely stabilized by the occluded conformation (Supplementary Table 4). The smaller dihedral angle of oxalate in the occluded conformation marginally stabilizes the oxalate binding, but the change is close to the margin of statistical error. The following sentences were added to the Discussion section.

(L.388-391, p.17)

“Furthermore, the calculated relative binding free energies of oxalate to OxIT revealed a significant stabilization in the occluded conformation compared to the outward-open conformation (i.e., ~20 kcal/mol decrease), which provides a physical basis for the conformational switch induced by oxalate binding (see Methods and Supplementary Table 4).”

4. With regard to the transport assay, a diagram and detailed explanation will help readers to understand this indirect assay. The western blot for all mutants and the co-expressed xenorhodopsin are needed to present.

On page 21, line 489, “the pH change”. Please define increase or decrease, and make this correction for all other places.

On page 21, line 476, “coupling with light-driven inward proton transfer”. Did you mean “the coupled light-driven inward proton transfer”? Can you detect format in the media?

Reply:

We added a schematic diagram of the transport assay to Figure 2d, as well as an explanation of the assay in the subsection “Oxalate-bound occluded structure” as described below. All the blots are displayed in Supplementary Figure 14.

We kept the word “change” in the phrase “the pH change” in the method section (on page 21, line 489 of the original manuscript) since we measured the change regardless of its direction (increase or decrease). Instead, we clarified that the pH change of interest is an increase in the same subsection in Results as described below.

As suggested by the reviewer, we also revised the phrase “coupling with light-driven inward proton transfer” in the sentence.

(L.202-205, p. 10)

“In the in cellulo transport assay, the extent of oxalate–formate exchange by OxIT in E. coli cells, which is negatively electrogenic, was assessed by coupled light-driven inward proton transfer by a co-expressed xenorhodopsin and the resultant pH increase of the external solution³⁶ (Fig. 2d).”

The formate concentration in the media resulting from the exchange with oxalate is estimated to be too low for detection (~ 16 nM vs. > ~ 40 μM for the lowest detection limit of a commercially available kit), making formate detection in the media unfeasible.

The data presented in the figure 2C show that most mutants have significant transport activity. With this indirect measurement on pH change, instead of the substrate of OxIT, and also only single time-point data from a 10-min incubation, it is not known what is the true mutational effect on the oxalate transport by OxIT. What is coupling ratio between H⁺ flow and oxalate transport?

Reply:

In order to examine the mutational effect further, we conducted two additional functional experiments: the GFP thermal shift assay to evaluate the ability of oxalate binding and the transport assay using the proteoliposomes reconstituted with the purified OxIT measured at multiple time points. The results are depicted in Figs. 2c and 2e.

We verified the mutational effects on transport activity reported in the original manuscript using a proteoliposome assay. Furthermore, by integrating the results from the three different functional assays, we expanded our discussion of the mutational effects as follows.

(L.190-202, p.9-10)

“Since the purified OxIT is unstable, particularly in the absence of substrate, we utilized a variety of functional assays: 1) the GFP thermal shift (GFP-TS) assay³⁵ to assess the ability of oxalate binding, utilizing the crude, detergent-solubilized OxIT-GFP fusion protein containing lipids from the host E. coli cells; 2) an in cellulo transport assay using E. coli cells recombinantly expressing OxIT³⁶; and 3) an in vitro transport assay using proteoliposomes

reconstituted with the purified OxIT. In the GFP-TS assay, wild-type OxIT exhibited thermal stabilization upon oxalate binding. In contrast, the extent of oxalate-dependent thermal stabilization was reduced in the mutant OxIT proteins Q34A, Y35A, Y124A, Y150A, N268A, R272A, W324A, and K355Q (Fig. 2c). Since the thermostabilities of the mutant OxIT proteins were not significantly different from that of the wild-type OxIT in the absence of oxalate (Supplementary Fig. 5), these results suggested that the mutation at the residues resulted in reduced binding affinity of oxalate and/or reduced stability of the bound structure.”

(L.207-221, p.10-11)

“Finally, the in vitro transport assay using the purified OxIT reconstituted into proteoliposomes was performed for R272A and the mutants which have not been tested by this method in the previous studies (Fig. 2e, Supplementary Fig. 6). A similar trend was observed between the proteoliposome and in cellulo assays, despite some small deviations due to the differences in the two systems. Intriguingly, while most mutants demonstrated a decrease in both binding and transport activities, Y124A and W324A mutations had no significant effects on transport activities, at least in either of transport assays, despite the reduced oxalate binding capacity. These observations may be explained by the fact that the lowering the affinity can result in acceleration of the net uptake due to rate-affinity trade-off and/or reduced substrate efflux³⁷. In contrast, Y328A and W352A showed similar oxalate binding but reduced oxalate uptake activities compared to the wild-type OxIT, indicating the importance of these residues on the catalytic turnover. Therefore, the results from the functional studies suggested that the residues in the vicinity of bound oxalate in the OxIT structure play significant roles in oxalate binding and/or transport.”

Please also see our reply to the first comment of Reviewer #2.

5. OxIT is a high-turnover number MFS transporter. Does the simple one-sidechain switching mechanism also contribute to the high-turnover number, in addition to the observed excess Gly residues in the protein?

Reply:

We appreciate the reviewer’s insightful suggestion regarding the structural basis for the high turnover observed on OxIT. We added the following discussion to the Discussion section.

(L.405-407, p.18)

“The mechanism of the conformational change that can be triggered by a single sidechain flip, along with the structural flexibility facilitated by the high glycine ratio, likely accounts for the rapid catalytic turnover exhibited by OxIT¹².”

Minor comments:

1) On page 16 line 367, add the OxIT source. Please clearly state from which bacterial strain the OxIT structure was made available.

Reply:

We have added the information.

2) On page 15, line 336, “allowing a spontaneous binding”. In fact, all binding events are spontaneous. I would delete “spontaneous” here and in other places.

Reply:

We appreciate the reviewer for pointing this out. We removed the word “spontaneous” throughout the manuscript.

3) Extended Data Table 2 needs an extensive table legend.

Reply:

We regret that the legend was insufficiently informative. The legend for the current Supplementary Table 3 has been updated as follows.

“The dihedral angle and binding distances of oxalate in the OxIT binding site are shown for the crystal structure and the optimized geometries from QM and QM/MM calculations. The results of the QM calculation with the nine frozen residues and free oxalate using either the B3LYP or B3LYP-D3BJ functionals are displayed. The results of the QM/MM calculations using the B3LYP-D3BJ functional with two different definitions of the QM region are also presented.”

4) O1-C1-C2-O2. Please define oxalate atom position in the crystal structure.

Reply:

We added a panel with the atom labels for the oxalate molecule as an inset of Figure 2a.

5) Label the helical number in the figure 2 DEF

Reply:

We added the labels for the helical numbers in the new Figs 2f, g, h.

6) Figure 4, the stick size is too big. It is difficult to view the substrate.

Reply:

We regret that the stick size makes it difficult to view the substrate. We reduced the stick size in Fig. 4.

7) In the Extended Data table 1 on the first row, add the important information (Apo and Oxalate-bound state, as well as their associated conformations). The R_{merge} values are too high since including the unused poor data beyond 3.0 Å resolution. The data quality that reflects the structure is not available. Please report the truncate the dataset to the refined levels.

Reply:

We have revised the first row in Supplementary Table 1.

When preparing the data for the refinement of the oxalate-bound form, we first performed the anisotropy correction of 2.6 Å data using the STARANISO server and the server determined the resolution cutoff as 2.6 Å after the correction. Then, we truncated the corrected data to 3.0 Å because the completeness of the data between 2.6 Å and 3.0 Å was very low. In this procedure, it is difficult to know the R_{merge} value for the data at 3.0 Å. Instead, we added the Supplementary Table 2, which displays the R_{merge} , $CC_{1/2}$, completeness, etc., per resolution shell. According to Supplementary Table 2, the R_{merge} value at 3.0 Å is 200–250. For reference, we performed the anisotropy correction on 3.0 Å pre-truncated data, and the server indicated that the R_{merge} value at 3.0 Å was 234.9, despite the resultant data being identical to our own. The high R_{merge} value was the results of merging data from numerous crystals to ensure accuracy of the diffraction intensity measurement.

8) Cite the pdb id in all figures and related text to distinguish the two crystal structures, and the crystal structure and MD simulations models.

Reply:

We cited the PDB identifiers at all pertinent locations in the manuscript.

Reviewer #2 (Remarks to the Author):

In this contribution by Jaunet-Lahary et al., the structure of an oxalate transporter in the presence and in the absence of a substrate is presented. The obtained resolution is much higher than previously reported (3 Å vs 6.5 Å) allowing assignment of side chains and importantly a substrate. I have no concerns about the structural work (apart from representation, see below), but my main concern is about the functional characterization of OxIT. The authors present it in a such minimal way that it looks suspicious. First of all it is absolutely indirect as they do not observe any transport of oxalate / formate, but monitor the acidification of the e.coli cells via inward proton pumping xenorhodopsin. The difference between the scenario (1) in the absence of oxalate, and the scenario (2) exposure to oxalate followed by illumination is not shown, and perhaps it is not even significant, and furthermore the internal buffer capacity of a cell is not taken into account and the impact of all other proton-transporting proteins is not considered. This experiment should be done preferably in vitro on OxIT reconstituted into proteoliposomes and in the best case with ¹⁴C-oxalate uptake assay for the direct visualisation of transport (which would also allow a proper characterisation of kinetics and not only speculation about it). In addition - affinity measurements are highly recommended, as so far there is only some qualitative indications (C2 binds stronger than C3 and C3 stronger than C4).

Reply:

We appreciate the reviewer's positive assessment of our structural study, and their insightful comments regarding the functional study.

According to the reviewer's suggestions, we have added two different kinds of functional assays, the results of which are shown in Figs 2c, 2e, Supplementary Figs 5, 6 and 7c.

Detergent-solubilized OxIT is found to be particularly unstable in the absence of substrate, a prerequisite for all ligand-related functional assays. To overcome the problem, we previously established the *E. coli* assay shown in Fig. 2C of the original manuscript and reported the details of the assay, including "the difference between the scenarios (1) in the absence of oxalate and (2) exposure to oxalate followed by illumination," in Hayashi *et al. Prot. Sci.* 2021 (ref. 36).

As suggested by the reviewer, we further performed the proteoliposome uptake assay. As depicted in Fig. 2e, a similar trend was observed between the proteoliposome and the original *E. coli* assays. The details of the validation of the analysis were summarized in Supplementary Fig. 6. We also kept the original *E. coli* assay results in Fig. 2d for the following reasons: 1) Since the detergent-solubilized and purified OxIT is unstable, we retained the *E. coli* assay, in which OxIT is never solubilized and has less chance of denaturation, to verify that the loss of activity observed in the proteoliposomes reconstituted with mutant OxITs is not attributed to

protein denaturation. 2) Unfortunately, during the revised experiment period, the reagent for the quantitation of the proteoliposome assay (e.g., oxalate oxidase and the accompanied kit) became unavailable and discontinued worldwide. Thus, we needed to limit the number of samples to be assessed with this method. We selected the mutants that had not been evaluated by proteoliposome assays in the previously reported studies among those tested in the *E. coli* assay.

Regarding the ligand affinity, we cited the previously reported K_d values for oxalate and malonate (C2 versus C3) as follows.

(L.229-231, p.11)

“This is consistent with the decreased affinity and transport activity for malonate with a K_d value of 1.2 mM compared to 0.02 mM for oxalate¹³.”

In addition, the GFP-TS assay was conducted with the crude, detergent-solubilized OxIT-GFP fusion protein containing lipids from the host *E. coli* cells. As demonstrated in Supplementary Fig. 7c, oxalate (C2) exhibited the highest specificity in comparison to malonate (C3) and succinate (C4).

In summary, we reinforced the functional characterization of OxIT, and the results are described on L.189–236, p. 9–11, in the subsection titled “Oxalate-bound occluded structure.” Please also see our response to reviewer 1’s comment number 4.

The authors put the great emphasis on the role of Q34, which they propose as a trigger to switch between open and occluded conformations. Although it seems feasible from their simulations, it is in a conflict with their functional data, where Q34A mutant retains at least 30% transport activity (that in principle should lead either to an arrest in a certain conformation, hence no transport, or deregulate the conformational change and lead to ~50% activity - should be possible to check with MD simulations on Q34A mutant).

Reply:

We appreciate the reviewer’s comment on this particular point. Based on the structure, the Q34A mutation is presumed to destabilize the occluded conformation rather than arrest OxIT in a certain conformation. Therefore, we believe that the result of partial loss of the activities is consistent with the structural observations. On this account, the Q34 sidechain interaction with K355 is more appropriately compared to a “latch of the periplasmic gate” rather than to a “switch of the transition from the occluded to the outward-facing conformations,” as we originally stated. It merely prevents a reverse process during the inward transport of oxalate, namely the backward transition from

the occluded to the outward-facing conformations. The subsection “Substrate-binding and conformational dynamics of OxIT” has been updated as follows.

(L.354-358, p. 16)

“These findings suggest that the Gln34 side chain, in conjunction with the hydrogen bond between Thr38 and Val240, functions as a “latch of the periplasmic gate” to prevent the transition from the occluded to the outward-facing conformations. Indeed, the Q34A mutant displayed a partial loss of the binding and transport activities relative to the wild-type protein (Fig. 2c–e), indicating that the mutation destabilizes of the occluded conformation.”

Based on the simulation results of this study, the transition from the occluded to the inward-open conformations is considered one of the rate-limiting steps of the OxIT transport, which determines the transport rate constant. Due to the limited time scale of our simulations, such a transition has never been observed, so we did not perform additional simulations in an attempt to quantify the decreased activities.

The fact of abundant Gly residues in this protein is noteworthy - and again the role of these Gly for proposed flexibility required for transport can be and should be checked via mutagenesis (in vitro and in silico)

Reply:

We appreciate the reviewer’s insightful comment. We concur that OxIT’s high glycine content in OxIT is essential for its transport activity and have added the following discussion to the Discussion section.

(L.405-407, p. 18)

“The mechanism of the conformational change that can be triggered by a single sidechain flip, along with the structural flexibility facilitated by the high glycine ratio, likely accounts for the rapid catalytic turnover exhibited by OxIT¹².”

Nevertheless, mutation studies for glycine residues, which are present at 52 positions in the OxIT molecule, and interpretation of the resulting data were anticipated to be too complex and were therefore omitted from this investigation.

Minor issues:

line 117: 1.5~1.6 Å should be ~1.5-1.6 Å

Reply:

We corrected this point.

line 124 and throughout the text: when you talk about certain residues it would be helpful to emphasise if they are conserved; these 52 glycines is actually the most likely reason why this protein is unstable without Fabs;

Reply:

We appreciate the reviewer's insightful comments. We added information regarding the conservation of glycine residues and its effect to the stability of OxIT to the subsection titled "OxIT structures in two different conformations" as follows.

(L.137-142, p.7-8)

"Conversely, the glycine-rich architecture is likely responsible for the instability of OxIT in detergent micelles, which hinders crystallization in the absence of antibody fragments and functional assays, as described below. The high glycine ratio was also observed in the other OFA proteins ($10.2 \pm 1.1\%$ with 15 strictly conserved positions; observed in 11 family members shown in Supplementary Fig. 3) and may be a family trait."

line 153 - it is not a substituting group, hence instead of carbamoyl should be carboxamide;

line 189 - supplanting? - replacement with?

Reply:

We corrected these points.

line 499 - western blots are mentioned, but not shown;

Reply:

We added Supplementary Fig. 14, displaying all Western blot results.

Fig 1. Positions of Gly can be moved to Supplementary;

Reply:

We removed the markings on the Gly positions in the panel D. The original panels have been transferred to Supplementary Fig. 2c.

Fig 2C - no negative control is shown, no statistical information given; 2B - the numbers of helices perhaps are not really necessary;

Reply:

We reposted the mock-transformed *E. coli* result from Hayashi *et al.* 2021 (ref. 36) in Fig. 2d as a negative control for comparison. We also added statistical information to

Figs. 2c–e. We eliminated the numbers of helices from Fig. 2b.

Supplementary Fig 3A - the colour of OMIT map should be changed (red is usually used for negative peaks); panel C - show the additional H-bond (change view);

Supplementary Fig 6 -should be enlarged, hard to see.

Reply:

We revised the new Supplementary Fig. 4 and 10 according to the reviewer's recommendations.

To summarise - it is a good structural work with a rather poor functional characterization and it is more suitable for a more specialised journal.

Reply:

Again, we appreciate the reviewer's favorable assessment of our structural work. Now we have improved the functional characterization, which was insufficient in the previous manuscript.

Reviewer #3 (Remarks to the Author):

The manuscript reports on crystal structures (~3Å resolution) of oxalate-bound and ligand-free bacterial oxalate transporter, Ox1T, in two conformations, occluded and outward-facing. The authors analyze the structural data and perform molecular dynamics (MD) simulations to conclude that two basic residues (Lys355 and Arg272) form salt bridges with the substrate, the dicarboxylate oxalate. The MD simulations show that the flip of a single side chain (Lys355) neighboring the substrate is sufficient to trigger the opening of the protein to allow substrate passage. The authors' results further support the notion of Ox1T functioning as an antiporter between oxalate and formate, rather than a uniporter of each molecule.

The structural data from x-ray crystallography provides the community with further insight into the architecture of Ox1T, and into the mechanism by which oxalate is recognized as a C2 (rather than C4) substrate. The authors describe how the conformational change between the occluded and outward-facing states is not simple a tilt of rigid helices but rather due to the flexible bending of transmembrane helices. A relatively high (12.4%) glycine content enables this plasticity.

The work is a contribution to the understanding of how Ox1T (and other transporters) can selectively bind substrates. That Ox1T functions as an antiporter is known, and this study shows atomic-level evidence of why this behavior is achieved.

Overall, the methodologies of crystallization, structure refinement, and simulation appear to be sound (standard), but the simulations cannot be reproduced as they are described in this manuscript (see comments below). The interpretation of the simulation results seems plausible, but the interpretation is not thoroughly supported by the reported results. The mechanism of substrate binding/unbinding, based on a single observed opening of a salt-bridge, is not rigorously proven or put into context of experimental evidence (kinetic assays). Additional evidence and experimental details are required.

The manuscript is clearly written but mixes present and past tense in many instances (see point 16 below). The discussion is nicely structured and easy to follow. The figures have been created with great care.

Below is a list of issues that must be clarified before being reviewed again for publication.

Reply:

We appreciate the reviewer's insightful comments, particularly regarding our simulation study. We revised the manuscript according to their recommendations. In addition,

experimental evidence was added to the revised manuscript.

1) Could the authors please clearly list all models built, e.g. Model1=(oxalate+protein, Lys355 protonated), Model2=(formate+protein, Lys355 protonated), etc.... and make a table of all trajectories that were simulated corresponding to each model, including total simulation time. If several independent trajectories were run for a single model, please state in the table.

Reply:

We have built four models (systems) with different combinations of the initial structure (outward-open or occluded structure), the protonated state of Lys355, and the bound ligand (oxalate or formate; in the case for the occluded structure). In Supplementary Table 5, we provide a summary of simulation systems as suggested.

The following sentences were added to the Method section.

(L.682-686, p.29)

“As a summary of the simulation systems, we have constructed four systems with different combinations of the initial structure (outward-open or occluded structure), the protonation state of Lys355, and the bound ligand (oxalate or formate; in the case for the occluded structure). In Supplementary Table 5, these simulation systems, as well as the number of trajectories and total simulation time, are summarized.”

2) Please explain clearly, for each trajectory, what the starting geometry is, and whether it corresponds to one of the models built.

Reply:

We also listed the starting structures in Supplementary Table 5.

3) For each binding event observed, where is oxalate at t=0? Where is formate at t=0? Figure 4B suggests with the arrow that at t=0 oxalate is outside of the protein and enters the space between the NTD and CTD.

Reply:

The binding events were observed in simulations initiated from the outward-open structure, which is not occupied by a preexisting ligand in the binding site. We performed two simulations beginning from the outward-open structure, with Lys355 protonated and deprotonated. The initial positions of the binding oxalate are outside of

the protein, as shown in the newly added Supplementary Figure 8. Consequently, they enter the binding site through the space between the NTD and CTD during simulations. In the case for formate, only simulations initiated from the occluded structure with formate already in the binding site were conducted, thus, no binding event was observed.

In the section entitled “Substrate-binding and conformational dynamics of OxIT,” we added the following sentence:

(L.294-296, p.14)

“We first simulated oxalate binding to the ligand-free outward-facing conformation (PDB ID 8HPJ) by positioning the oxalate outside of the protein (Fig. 4,a–c, Supplementary Fig. 8).”

4) The authors write “several binding and unbinding events were observed for the neutral Lys355” or “Spontaneous binding of oxalate” was observed, but they never clearly define what a “binding” or “unbinding” event is. How is “binding” defined? What is a “single binding event”?

Reply:

We regret that the definition of binding was ambiguous. Binding and unbinding events are defined by the distance between the geometric centers of the oxalate ion and the binding site residues (Q34, Y35, Y124, Y150, R272, W324, Y328, W352, K355). As shown in Fig. 4c, a binding event is defined by a distance of less than 5 Å because the oxalate ion clearly interacts with the binding site residues with this distance. An unbinding event is defined by the same distance greater than 5 Å. A single binding event refers to the continuous binding of the same oxalate ion, as observed for the protonated Lys355, whereas multiple binding and unbinding of different oxalate ions were observed for the neutral Lys355, as shown in different colors in Fig. 4c.

We added the following sentence to the corresponding section:

(L.297-298, p. 14)

“The distance between the geometric centers of the oxalate ion and the binding site residues, with a cutoff distance of 5 Å, determined the binding (Fig. 4c).”

We also altered the following phrase.

(L. 305-308, p. 14)

“For the protonated Lys355, a single binding event was observed, and the bound oxalate ion mostly remained in the binding site for the remainder of the simulation (grey line in the top panel of Fig. 4c). In contrast, for the neutral Lys355, multiple binding and unbinding events of

different oxalate ions were observed (coloured lines in the bottom panel of Fig. 4c)."

5) How long does the substrate occupy the "binding position"? In other words, what is the occupancy rate?

Reply:

In the case for the model with protonated Lys355, oxalate entered the binding site at the onset of the simulation and mostly remained there for the duration of the simulation. Thus, the occupancy rate calculated by the above definition using the distance between the oxalate and binding site is 98.6%. In contrast, in the case for the model with deprotonated Lys355, multiple unbinding and rebinding events of different oxalate ions were observed, resulting in a lower occupancy rate of 77.0%. In theory, the occupancy rate is related to the dissociation constant, which is difficult to accurately estimate and beyond the scope of this study due to the limited sampling of binding and unbinding events.

In the section entitled "Substrate-binding and conformational dynamics of OxIT," we added the following sentence:

(L. 308-312, p. 14)

"This results in a higher occupancy rate of 98.6% for the protonated Lys355 than that of 77.0% for the neutral Lys355, which is calculated by the above definition of the bound state using the oxalate and binding site distance. Thus, a lower oxalate dissociation constant is predicted for the protonated Lys355."

6) Figure 4C is of little help to understand binding. What exactly is being plotted as a function of time? Colors refer to different oxalate molecules, but how many independent trajectories are presented? How many oxalate molecules are present per model? What is the meaning of the distance signal >10Å not shown?

Reply:

We regret that the figure might be confusing. Each panel of Fig. 4c represents a single trajectory with protonated or deprotonated Lys355, plotting the distances of all 58 oxalate ions included in the simulation from the binding site. The distance was calculated using the geometric centers of the oxalate ion and the nine residues in the binding site (Q34, Y35, Y124, Y150, R272, W324, Y328, W352, K355). In the revised Fig. 4c, the distances up to 16 Å are displayed. Since the distance threshold to discuss binding/unbinding events is defined as 5 Å as described above, we focused the events

occurring within 16 Å for clarity.

We also modified the Figure legend as follows:

(L. 1019-1021, p.40-41)

“Distances of the oxalate ions from the binding site in a single trajectory, either with protonated Lys355 (top) or deprotonated Lys355 (bottom), are shown. The geometric centers of the oxalate ion and the binding site residues were used to calculate the distance. Different colors represent the different oxalate ions included in the simulation.”

7) Is the flip of Lys355 side-chain due to charge repulsion with side chain of Arg272 a simulation event that can be reproduced? What happens if Lys355 is mutated to Ala? What happens if Arg272 is mutated to Ala?

Reply:

In simulations initiated from the occluded conformation with oxalate or formate, the side-chain flip of Lys355 was observed in the trajectories with opening the pathway to outside, as similar to the ligand-free outward-facing crystal structure.

(Left) A snapshot at 580 ns in a simulation initiated from the occluded state with oxalate in the binding site. K355 occasionally flipped, after OxIT changed its conformation to the outward-open state, that is, after the binding site was filled with water molecules from outside (Fig. 4f). Oxalate moves away from R272 and K355 side chain flips toward Q63. (Right) A snapshot at 665 ns in a simulation initiated from the occluded state with formate in the binding site. The same situation was observed. Water molecules are not shown for clarity.

Conversely, in simulations initiated from the outward-open conformation, the Lys355 side chain was observed to restore the configuration from the flipped position to that observed in the oxalate-bound occluded crystal structure, by interacting with the ligand. Therefore, the simulations reproduced a switchable behavior of the side-chain configuration of the Lys355.

In the section entitled “Substrate-binding and conformational dynamics of OxIT,” we added the following sentence:

(L. 468-469, p.14)

“The interaction of oxalate ion with Lys355 resolved its charge repulsion with Arg272 in the ligand-free form and restored the side chain’s configuration from the flipped state.”

The mutation effects of Arg272 or Lys355 have been experimentally investigated and observed to render the transporter inactive, as depicted in Fig. 2c–e.

8) What experimental kinetic assays (literature?) have been performed that may give us an idea of the reaction rates of oxalate binding/unbinding? Can the observed “binding event” be related to DeltaG via transition state theory to experimental evidence?

Reply:

Previously, transport rates of OxIT have been determined by oxalate uptake assays using reconstituted proteoliposomes [Anantharam *et al.* 1989 (Ref. 11), Ruan *et al.* 1992 (Ref. 12), Abe *et al.* 1996 (Ref. 46)]. However, to our knowledge, the oxalate binding/unbinding rates have never been experimentally addressed. As we have seen in this study, conformational transitions of OxIT and not oxalate binding are likely the rate-limiting steps. Therefore, it is not feasible to deduce the oxalate binding/unbinding rates from the reported transport rates.

It is also difficult to calculate DeltaG from simulations due to the limited sampling of binding/unbinding events. To address this issue, we have calculated the relative binding free energies of oxalate to OxIT using the MM-GBSA method for the occluded and outward–open conformations (see Methods). We found that the occluded conformation stabilizes the binding significantly more than the outward-open conformation (Supplementary Table 4). This result provides a physical explanation for the conformational change of OxIT induced by oxalate binding. Please also see our responses to the comment number 3 from Reviewer 1.

9) The authors write that the binding event changes depending on the protonation state of

Lys355. The authors also mention pH of the gut, a macroenvironment, but the pH of the gut is likely not the explanation for the deprotonation of a sidechain buried in a protein. Perhaps the authors should discuss how the protonation of Lys in buried hydrophobic pockets can change. Please check Takayama et al. “Direct Evidence for Deprotonation of a Lysine Side Chain Buried in the Hydrophobic Core of a Protein” in JACS 2008.

Reply:

We wish to thank the reviewer for their suggestion. We concur that a hydrophobic environment can influence the protonation state of Lys as described in Takayama *et al.* JACS 2008. This possibility was added to the corresponding sentence as follows:

(L. 302-305, p.14)

“The stability of the bound conformation depended on the protonation state of Lys355 (see Methods for pK_a calculation), which can be influenced by the luminal pH in the gut, which varies from 5 to 8 by region ⁴⁰, and the hydrophobic environment in the binding site ⁴¹.”

10) The authors seem to confuse atomic charges and simulation parameters in the discussion of QM and QM/MM. How are charges for the small molecules calculated? How are parameters (bonding and non-bonding) determined for these substrates? Have these charges and parameters been validated and tested? Are they robust? Are any other simulations involving oxalate or formate known? Given that the mechanism of binding relies heavily on the charges and parameters of the substrates, the quality of these computed numbers is significant for interpretation of results.

Reply:

The parameters and charges used for the oxalate ligand are based on the researches of Kroutil *et al.* (ref 43 and 75), as described in the Method section. In their works, force-field parametrization was performed because oxalate is a non-standard ligand due to the proximity and electrostatic repulsion between two carboxyl groups.

The atomic charges of oxalate complexed with OxIT were calculated using the Restrained Electrostatic Potential (RESP) scheme, which is commonly used for MM parametrization of non-standard ligands with the AMBER force field. Moreover, a global charge of $-2e$ is ensured for the ligand within the OxIT protein. Concerning the solvated oxalate molecules, the Electronic Continuum correction with Rescaling (ECCR) is applied to the atomic charges to achieve a good agreement with Ab Initio Molecular Dynamics (AIMD) results, particularly for the radial distribution functions involving water molecules. The bonding and non-bonding parameters were derived from the Amber99SB force field parameters with minor adjustments based on the AIMD results

(ref 43 and 75).

Several MD studies involving oxalate are present in the literature. These studies have simulated the solvation of oxalate anion in bulk, its complexation with calcium cation, and the adsorption process on the surface. However, to our knowledge, there is no study that simulates oxalate complexed with protein. Consequently, deciding which oxalate model to employ for this purpose is a difficult endeavor. In this study, we have decided to adopt the oxalate model with the RESP charges.

We added and the following sentences with additional references to the Methods section:

(L.662-664, p.28)

“To our knowledge, no study has yet simulated oxalate complexed with protein, although several MD studies have simulated the solvation of oxalate anion in bulk, its complexation with calcium cation, and the adsorption surface process⁷⁸⁻⁸⁰.”

11) What is the “original RESP” scheme? What software was used?

Reply:

In response to this comment, the following sentences have been modified and added to the Methods section:

(L. 658-662, p. 28)

“The oxalate ligand in the binding site of OxIT was described using parameters determined by the Restrained Electrostatic Potential (RESP) scheme⁷⁶ without applying the ECCR correction, considering that the protein environment differs from that of water solution. The RESP charges have been calculated by the Antechamber software⁷⁷.”

12) Could the authors please comment on the choice of the membrane composition? Why was a homogenous membrane (phosphatidylethanolamine) chosen?

Reply:

In response to this suggestion, the following sentences were added to the Methods section:

(L.646-649, p. 28)

*“The PE lipid is a major component in both *O. formigenes*⁶⁹, from which OxIT is derived, and *E. coli*⁷⁰, in which transport assays were conducted. In addition, there is no evidence that other specific lipids are required for OxIT activity.”*

13) What are the x, y, z, dimensions of the final system? Only x,y dimensions of bilayer are given.

Reply:

In response to this comment, the following sentence was modified:

(L.649-650, p. 28)

“The protein–membrane system was solvated with TIP3P water molecules and 150 mM KCl, resulting in the z dimension length of 100 Å.”

14) How many Cl⁻ ions are replaced in each model with oxalate ions? How is the model containing formate built if formate is constructed at one site (line 533)?

Reply:

In response to these remarks, the following sentences were added to the Methods section:

(L.650-652, p. 28)

“Then, all 87 Cl⁻ were replaced with 58 oxalate ions using AmberTools17⁷¹ without altering the total charge by taking into account the scaled effective charge (−1.5e) of the oxalate model in solution (see ECCR below).”

(L.676-678, p. 29)

“In addition, a K⁺ ion was removed from the prior model and replaced with a water molecule to account for the loss of a negative charge.”

15) What is the integration time step (1 fs? 2 fs?) in the simulations?

Reply:

In response to this comment, the following sentence was added to the Methods section:

(L.673, p.29)

“The integration time step was 2 fs.”

16) Please stay in present tense when discussing results (e.g. line 111 should be “molecule displays”, line 112 should be “which is connected”, line 119 should be “are observed”, line 328 “results suggest”).

Reply:

We corrected all these phrases.

17) Lines 137-138 verb is missing, probably should read “is refined”?

Reply:

We rectified this.

18) What does “optimised” mean in this context? Structural refinement?

Reply:

The corresponding sentence was revised as follows:

(L.720-721, p. 31)

“As with QM/MM calculations, optimised structures, obtained as stationary points on the potential energy surface, were true energetical minima without imaginary frequencies.”

19) Lines 210-212 verb is missing

20) Line 278 should be “consisting of Thr38”

21) Lines 321-322 “change” and “changes” too close together

22) Line 532 should read “pointing toward Lys355” or similar

Reply:

We corrected each of these issues.

REVIEWERS' COMMENTS

Reviewer #1 (Remarks to the Author):

The authors took my comments seriously and provided new data which supported and extended the original conclusion. I accept this version for publication.

Reviewer #2 (Remarks to the Author):

I thank the authors for a substantial revision of their original manuscript. Most of the issues have been successfully resolved and now the authors provide some functional characterisation. I do recognise the efforts especially since the protein is rather unstable in the apo form, and the disappearance of certain reagents is also something beyond our control. With these new data obtained, the Y124A behaviour is quite interesting - it looks almost as if the gain-of-function mutant. How do the authors reconcile the functional data for this mutant and the fact that Y124 is one of the key residues in the binding site with the direct interaction with the substrate?

The fact that the authors used only PE for their MD simulations is indeed somewhat puzzling - I do agree with the notion that PE is the major component (~60% in E.coli lipid extract), but normally simulations are done as a mixture of PE and PG. Furthermore, for reasons of transparency and reproducibility, please upload MD-related data to Zenodo repository.

Minor issues:

line 267 - instead of 'However that between Lys355...' perhaps better to say 'However, the one between Lys355...'

Line 425 - dodecylmaltoside is misspelled - also mention at the first instance whether it is alpha or beta-DDM

Line 981 - looks like the word 'was' is not necessary here

Lines 991-92 I am not entirely sure it is a valid way to include these 'mock' data from the previous study, may look too confusing, perhaps just refer to it in the figure legend

Line 995 - I cannot see 'empty', do you mean 'mock'?

Line 1009 - give the values of potential map (in kT/e-)

For figure 4 - it would be nice if a movie is included here as well.

Reviewer #3 (Remarks to the Author):

The authors have addressed all of my comments and I have no further issues with the manuscript.

Responses to reviewers

Reviewer #1 (Remarks to the Author):

The authors took my comments seriously and provided new data which supported and extended the original conclusion. I accept this version for publication.

Reply:

The authors thank again the reviewer's previous insightful suggestions, which helped us revise and improve our manuscript. We are happy to know that now the reviewer has accepted it for publication.

Reviewer #2 (Remarks to the Author):

I thank the authors for a substantial revision of their original manuscript. Most of the issues have been successfully resolved and now the authors provide some functional characterisation. I do recognise the efforts especially since the protein is rather unstable in the apo form, and the disappearance of certain reagents is also something beyond our control.

Reply:

The authors thank again the reviewer's previous insightful comments and suggestions, and their kind understanding of our efforts in the additional experiments. We also appreciate their careful reading and insightful comments on our revised manuscript. Based on their feedback, we have further revised our manuscript as follows.

With these new data obtained, the Y124A behaviour is quite interesting - it looks almost as if the gain-of-function mutant. How do the authors reconcile the functional data for this mutant and the fact that Y124 is one of the key residues in the binding site with the direct interaction with the substrate?

Reply:

We appreciate the reviewer's thoughtful comments. We had a general discussion about the rate-affinity trade-off and the reduced substrate efflux in L.207-211, p. 9-10 in the subsection entitled "Oxalate-bound occluded structure" in the Results section. Furthermore, especially for the particular mutant noted by the reviewer, we added a discussion in terms of both the structure and transport kinetics in the "Substrate-binding and conformational dynamics of OxIT" subsection as follows.

(L.374-378, p. 16)

“One of the enigmatic mutants showing reduced oxalate binding but retained transport activity, Y124A (Fig. 2c-e), locates at the entrance of the cytoplasmic gate underneath the bound oxalate (Figs. 2f and 4f). The mutation might destabilize the hydrophobic layer at the gate and facilitate its opening, probably the rate-limiting step of the transport, and thus could compensate for the reduced affinity to oxalate in its transport activity.”

The fact that the authors used only PE for their MD simulations is indeed somewhat puzzling - I do agree with the notion that PE is the major component (~60% in E.coli lipid extract), but normally simulations are done as a mixture of PE and PG.

Reply:

According to reference #71 (Chamberlain *et al. Anal Bioanal Chem* 2019), the content of PE and PE-derivative lipids in *O. formigenes* is more than 80%, while that of PG is less than 3%. Therefore, we believe that the conditions for our MD simulations resemble a physiological situation in *O. formigenes* and thus are appropriate.

Furthermore, for reasons of transparency and reproducibility, please upload MD-related data to Zenodo repository.

Reply:

We appreciate the reviewer's important advice. We uploaded all the MD related data to Zenodo and provided the DOI in the "Data availability" section as follows.

(L.740-741, p. 31)

“The MD-related data have been deposited in the Zenodo repository [<https://doi.org/10.5281/zenodo.7597686>] ⁹⁴.”

Minor issues:

line 267 - instead of 'However that between Lys355...' perhaps better to say 'However, the one between Lys355...'

Line 425 - dodecylmaltoside is misspelled - also mention at the first instance whether it is alpha or beta-DDM

Line 981 - looks like the word 'was' is not necessary here

Reply: We corrected the points on lines 267, 425, and 981, which are on lines 262, 432, and 1021 in the revised manuscript, as suggested by the reviewer.

Lines 991-92 I am not entirely sure it is a valid way to include these 'mock' data from the previous study, may look too confusing, perhaps just refer to it in the figure legend

Reply:

We appreciate this suggestion. We removed the ‘mock’ data from Figure 2d and added the description about the “mock” data, which were collected from the *E. coli* cells harboring solely xenorhodopin without OxIT and set as the background (0%) in the analysis, in the figure legend as follows.

(L.1028-1030, p. 39)

“The relative transport activities of the mutant OxIT to that of the WT OxIT measured on the same day set as 100%, while that of the cells without expressing OxIT³⁶ was set as 0%, are displayed.”

Line 995 - I cannot see 'empty', do you mean 'mock'?

Reply:

We appreciate the reviewer’s this pointing. The label “mock” in Figure 2e was replaced with “empty.”

Line 1009 - give the values of potential map (in kT/e-)

Reply:

We added this information on line 1050 in the revised manuscript.

For figure 4 - it would be nice if a movie is included here as well.

Reply:

We appreciate the reviewer’s valuable suggestion. We added Supplementary Movies 1 and 2, which are related to Fig. 4e and 4f, as follows.

*“**Supplementary Movie 1.** The first 600 ns trajectory of OxIT starting from the occluded conformation, which remained in the occluded state, is shown. The representation is the same as Fig. 4e, except all water molecules are shown in the CPK color.*

***Supplementary Movie 2.** The first 500 ns trajectory of OxIT starting from the occluded conformation with a transition to the outward-open state is shown. The representation is the same as Fig. 4f, except all water molecules are shown in the CPK color.”*

Reviewer #3 (Remarks to the Author):

The authors have addressed all of my comments and I have no further issues with the manuscript.

Reply:

The authors thank again the reviewer's previous valuable suggestions. Their comments assisted us in revising and improving our manuscript.